# MERGEMIX: A UNIFIED AUGMENTATION PARADIGM FOR VISUAL AND MULTI-MODAL UNDERSTANDING

**Xin Jin**[1,*]   **Siyuan Li**[1,2,*]   **Siyong Jian**[1]   **Kai Yu**[1]   **Huan Wang**[1,†]

[1]Westlake University, Hangzhou, China
[2]Zhejiang University, College of Computer Science and Technology, Hangzhou, China
{jinxin86; lisiyuan; wanghuan}@westlake.edu.cn
[*] Equal contribution    † Corresponding author
https://github.com/JinXins/MergeMix

## ABSTRACT

Vision-language alignment in multi-modal large language models (MLLMs) relies on supervised fine-tuning (SFT) or reinforcement learning (RL). To align multi-modal large language models (MLLMs) in the post-training stage, supervised fine-tuning (SFT) is a stable choice but requires human annotations and lacks task generalizations, while Reinforcement Learning (RL) searches for better answers from reward signals but suffers from computational overhead and instability. To achieve balance among scalability, efficiency, and alignment generalizations, we propose MergeMix, a unified paradigm that bridges SFT and RL with an efficient Token **Merge** based **Mix**up augmentation. As for the Mixup policy, we generate contextual aligned mixed images with the corresponding labels according to the merged attention maps with cluster regions. Then, we enhance the preference-driven paradigm for MLLMs by building preference pairs with raw images and MergeMix-generated ones and optimizing the soft preference margin with the mixed SimPO loss. Extensive experiments demonstrate that MergeMix not only achieves dominant classification accuracy as an augmentation method but also improves generalization abilities and alignment of MLLMs, providing a new learning paradigm for preference alignment with training efficiency and stability.

## 1 INTRODUCTION

Multi-modal Large Language Models (MLLMs) (Liu et al., 2024b; Bai et al., 2025; Tong et al., 2024) have recently demonstrated remarkable capabilities in integrating visual and textual information, enabling a wide range of applications from visual question answering to multi-modal reasoning. Since MLLMs are typically pre-trained on massive web-scale datasets, forcing them to possess a wide range of knowledge and general reasoning capabilities, Supervised Fine-Tuning (SFT) and Reinforcement Learning (RL)-based preference optimization (Yang et al., 2025c) have emerged as two primary paradigms for aligning MLLMs with human preferences and specific task requirements. However, SFT depends on high-quality instruction–response annotations and optimizes the likelihood of reference responses, which does not explicitly model relative preferences between outputs. RL-based methods such as RLHF are more preference-aware, but require an additional reward model that may introduce bias or be exploited by the reward signal.

Due to the shortcomings of data quality and training efficiency, some works (Zhu et al., 2024; 2025b; Luo et al., 2024; Tan et al., 2025; Wang et al., 2024b) try to build performance pairs for optimization. How to build the preference pair with control and high-quality data for model training is the remaining open question. For example, SeVa (Zhu et al., 2024) proposed a preference optimization method by building a loser through some classic augmentation (*i.e.*, RandomCrop). Then, select the different responses for optimizing the model by a DPO loss (Rafailov et al., 2023). However, these methods have two drawbacks: the augmentations are highly random, and the DPO loss cannot be related to the data, which means SeVa can only select useful training data. Those technical causes SeVa can not control the quality of the loser, which is harmful for some visual question answering tasks, and reduces the training data by selecting "hard negatives". Hence, we investigate an inter-

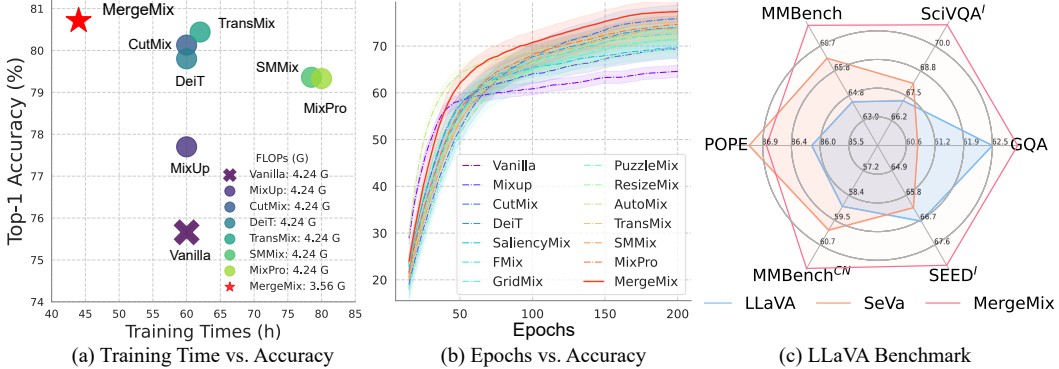

Figure 1: **Efficiency and performance for MergeMix.** (a) The training time *vs.* accuracy of mixup methods with the DeiT-Small model. (b) The image classification Top-1 accuracy *vs.* training epochs of different mixup methods on the CIFAR100 dataset with the DeiT-Tiny model. (c) The radar plot of the results on part VQA tasks by LLaVA-7B, LLaVA with SFT, and MergeMix.

esting question: *Is it necessary to propose novel techniques rather than some classical machine learning methods in the MLLM scenario?*

In this paper, we revisit the mixup augmentation, which synthesizes mixed samples and corresponding labels with given mixing ratios. However, two main challenges arise as illustrated in Figure 1: (1) achieving an optimal trade-off between efficiency and performance of mixup augmentations that rely on saliency-based metrics, (2) extending the augmentation to MLLMs properly, from classical image corruptions to data-dependent samples. Motivated by these perspectives, we propose a novel training framework called **MergeMix**, which builds preference pairs for MLLM training through data augmentation methods and ranking loss, thereby bridging the gap between SFT and RL. Figure 2 shows the two scenarios of MergeMix. **(a)** We introduce MergeMix, a novel data augmentation that generates mixed samples through token merge techniques. A bipartite soft matching strategy captures similarity information that preserves contextual features, ensuring the mask retains useful information. Meanwhile, MergeMix links the merge ratio and mixing ratio, aligning mixed images with the corresponding labels, enabling precise mixing data generation. **(b)** We propose a preference-driven paradigm for MLLMs, where augmented samples are defined as non-preferred responses (Loser) and clean samples as preferred responses (Winner). This paradigm facilitates preference tuning via the mixed SimPO loss, and leverages the mixing ratio as the soft preference margin to enable adaptive optimization. Altogether, Figure 1 shows these contributions yield an efficient and effective training strategy that achieves stronger alignment with human preferences while preserving the stability and scalability of SFT. Since the optimization object has a direct relationship with augmentation, it obtains a more robust ability in calibration. Extensive experiments show that MergeMix, as a training-time augmentation paradigm, achieves competitive performance in both image classification and MLLM benchmark with favorable efficiency.

Our contributions can be summarized as:

(a) We use token merging to obtain a local clustered attention map, enabling the generation of mixed images with cluster regions, a label re-scaling strategy aligned the mixed images with their corresponding labels, achieve well performance on both overhead and classification accuracy.

(b) We enhance the preference tuning paradigm for supervised fine-tuning of MLLMs, where mixed images are treated as losers, the mixing ratio is used as a soft preference reward score, and optimize the model adaptively via the mixed SimPO loss.

(c) We validate that our method achieves state-of-the-art on several image classification datasets and benchmarks, along with the advantages of our training paradigm on several MLLM benchmarks.

## 2 RELATED WORK

In this section, we introduce the existing mixup approaches for image classification and token compression approaches in multi-modal large language models for efficient training or inference.

**Mixup Augmentations**   The Mixup method mitigates model overfitting by generating augmented samples through mixing two different images within a mini-batch. Broadly, Mixup methods can be categorized into two types: **Static**, which relies on human priors or randomness, and **Adaptive**, a data-dependent type that leverages certain metrics to guide the mixing process. **(i) Static:** MixUp (Zhang et al., 2017) generates mixed samples via linear interpolation with $\lambda$. CutMix (Yun et al., 2019) extends this idea from the global pixel level to a local patch level by constructing a mask of size proportional to $\lambda$ to mix images. ResizeMix (Qin et al., 2020) ensures that features from at least one class are always preserved in the mixed sample by resizing the source image before mixing. Other methods, *e.g.*, FMix (Harris et al., 2020), SmoothMix (Lee et al., 2020), GridMix (Baek et al., 2021), and StarMix (Jin et al., 2024b), focus on improving the mask to obtain more suitable mixed samples. **(ii) Adaptive:** SaliencyMix (Uddin et al., 2021) employs a saliency extractor to identify informative patches in images for mixing. Attentive-CutMix (Walawalkar et al., 2020) and Super-Mix (Dabouei et al., 2021) utilize a teacher model to guide mask generation. PuzzleMix (Kim et al., 2020) and Co-Mix (Kim et al., 2021) generate appropriate masks based on gradient information obtained from a forward pass of the samples. AutoMix (Liu et al., 2022) and AdAutoMix (Qin et al., 2024a) adopt an end-to-end bi-optimization paradigm to produce mixed samples. TransMix (Chen et al., 2022), SMMix (Chen et al., 2023), and MixPro (Zhao et al., 2023) specifically enhance ViTs by computing attention scores from a forward pass to generate feature-aware masks and further refine the label ratio through attention scores. DiffuseMix (Islam et al., 2024a) and GenMix (Islam et al., 2024b) generate mixed samples by a diffusion model (Zhu et al., 2025a) for label preserving.

**Token Compression in MLLMs**   Token compression methods (Bolya et al., 2023; Tao et al., 2025a; Shao et al., 2025; Chen et al., 2025; Tao et al., 2025b) mainly propose to merge or drop redundant tokens to achieve efficiency and acceleration. In MLLMs, images and texts will incur a significant number of tokens, which are often full of redundant information. Obvious researchers bring the token compression into MLLMs. Overall, we divide the methods that reduce tokens into 2 types, **Reduce in Encoder** and **Reduce in Decoder**. **(i) Reduce in Encoder:** MADTP (Cao et al., 2024) aims to achieve MLLM acceleration by purging visual tokens. LLaVA-PruMerge (Shang et al., 2024) uses the attention of `[CLS]` token to select clustering centers and then merges the remaining tokens with lower attention through a KNN clustering and weighted clustering center updating mechanism. VisionZip (Yang et al., 2025b), instead, retains visual tokens with high attention scores and subsequently merges the remaining tokens through clustering. Others, such as TokenPacker (Li et al., 2025), AVG-LLaVA (Lan et al., 2025), MustDrop (Liu et al., 2024c), and LLaVolta (Chen et al., 2024a), achieve acceleration by choosing a metric to sample TopK visual tokens. FastVLM (Vasu et al., 2025) proposes an Efficient Vision Encoder to reduce visual tokens. **(ii) Reduce in Deocder:** PyramidDrop (Xing et al., 2024) divides the token compression process in LLM into multiple stages, which employs a pyramidal token drop to avoid losing too much visual information in shallower layers. ATP-LLaVA (Ye et al., 2025) proposes an Adaptive Token Pruning (ATP) module that reduces the number of tokens in the decoder layer. ZipVL (He et al., 2024) proposes a dynamic ratio allocation strategy via the importance token, adaptively determined based on the distribution of attention scores in a particular layer, rather than a fixed hyperparameter.

## 3   PRELIMINARIES

**Reformulation of Mixup Augmentation.**   We define $\mathbb{X}$ to be the set of training samples and $\mathbb{Y}$ the set of ground truth of the corresponding labels. For each sample pair $(x, y)$, we randomly sample two pairs in $\mathbb{X}$ and $\mathbb{Y}$, with $\lambda$ in $\text{Beta}(\alpha, \alpha)$. The mixed images and labels are generated by applying the optimized mask $\mathcal{M}$ and ratio $\hat{\lambda}$, which come from a defined policy $\mathcal{P}(\cdot, \cdot, \cdot)$ according to Eq. (1):

$$\mathcal{M}, \ \hat{\lambda} = \mathcal{P}\big(f_\theta(x_i, x_j), \ (y_i, y_j), \ \lambda\big), \tag{1}$$

$$\begin{aligned} \hat{x} &= \mathcal{M} \odot x_i + (1 - \mathcal{M}) \odot x_j, \\ \hat{y} &= \hat{\lambda} * y_i + (1 - \hat{\lambda}) * y_j, \end{aligned} \tag{2}$$

where the $\odot$ denotes element-wise multiplication. Policy $\mathcal{P}(\cdot, \cdot, \cdot)$ aims for the $\mathcal{M}$ to retain more features in the mixed sample. The $\hat{\lambda}$ keeps the initial sampling ratio when without optimization; otherwise, the $\lambda$ can be re-computed by some metrics (*i.e.*, MergeMix uses the total mask values).

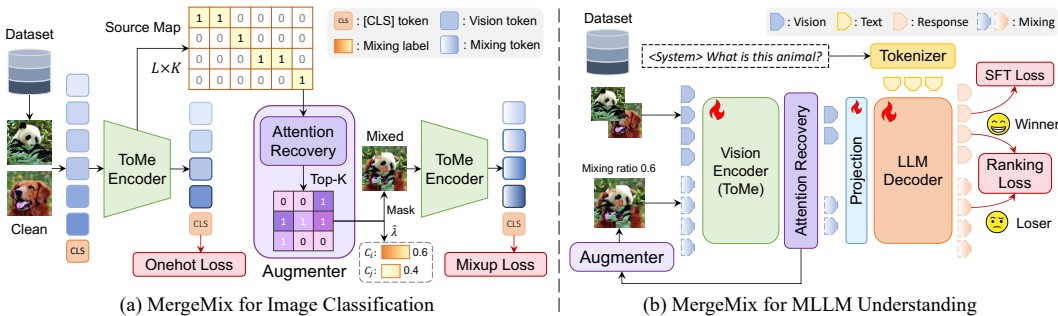

Figure 2: **Overview of MergeMix in two scenarios. (a) MergeMix for Image Classification:** The image is processed by the ToMe encoder, with Attention Score Recovery and TopK sampling to generate the corresponding class prediction. **(b) MergeMix for MLLM:** Preference pairs are encoded by the vision model with token merging, and the LLM decoder generates response text for the loser and winner, optimized via a ranking loss.

**Preference Tuning for MLLMs.** Preference optimization methods aim to align LLMs and MLLMs with human feedback by contrasting preferred and dispreferred responses. A general preference loss can be abstractly defined as Eq. (3):

$$\mathcal{L}_{\text{Pref}} = -\log \sigma \big( \pi_\theta(x, y^+) - \pi_\theta(x, y^-) \big), \tag{3}$$

where $(x, y^+)$ and $(x, y^-)$ denote the preferred and dispreferred responses respectively, and $s_\theta$ is a scoring function that reflects model preference. Different approaches (*e.g.*, PPO (Schulman et al., 2017), DPO (Rafailov et al., 2023)) instantiate $\pi_\theta$ in various ways, but all share the same principle.

Unlike RL-based approaches, which require training a separate reward model, DPO provides a simple and stable alternative by directly optimizing the policy model using preference pairs. Formally, the DPO loss is defined as Eq. (4):

$$\mathcal{L}_{\text{DPO}} = -\mathbb{E}(x, y^+, y^-) \sim \mathcal{D} \Big[ \log \sigma \big( \beta \log \frac{\pi_\theta(y^+|x)}{\pi_{\text{ref}}(y^+|x)} - \beta \log \frac{\pi_\theta(y^-|x)}{\pi_{\text{ref}}(y^-|x)} \big) \Big], \tag{4}$$

where $\pi_\theta$ denotes the policy model (in our case, an MLLM such as LLaVA-v1.5 (Liu et al., 2024b) or Qwen-VL (Bai et al., 2025)), and $\pi_{\text{ref}}$ represents a frozen reference model used to preserve alignment with the original pre-trained distribution. $\sigma$ is the sigmoid function, and $\beta > 0$ is a temperature-like scaling factor that controls the sharpness of preference separation. Intuitively, DPO encourages the policy to assign a higher likelihood to preferred responses $(y^+)$ than to non-preferred ones $(y^-)$, while maintaining proximity to the reference model.

## 4 MergeMix Training Paradigm

In this section, we present the implementation of MergeMix, an augmentation approach via token merging for image mixing, not only for image classification, but also designed for multi-modal large language models. Figure 3 shows the overall pipeline of MergeMix, and we describe in two subsections in detail, which are from the input space to the loss objective for model training.

### 4.1 Image Mixing via Token Merge

In MergeMix, we leverage the relationship between the merge ratio and mixing ratio. The merge ratio measures the information of raw samples, while the mixing ratio balances the information between mixing samples, thereby enabling precise data generation of mixed inputs and labels. In this subsection, we first introduce MergeMix on the input space. Then we use the designed mixing policy $\mathcal{P}(\cdot, \cdot, \cdot)$ to obtain the mixed images $\hat{x}$ with the mask $\mathcal{M}$.

**Image Policy with Token Merging.** Unlike other mixup methods (Chen et al., 2022; 2023; Zhao et al., 2023), we introduce a ViT-based model $f_\theta(\cdot)$ iteratively replace $N$ attention layers with ToMeAttention as ToMe (Bolya et al., 2023), Given the initial sequence $Z_L = f_\theta(\hat{x})$, then merges tokens as Eq. (5):

$$S, A_K, Z_K = \text{ToMeAttention}(Z_L, r), \tag{5}$$

where $A_K$ denotes the attention map from the model, and $Z_K$ denotes the feature tokens for computing one-hot loss. $r$ denotes the number of merged tokens, which can reduce some high-similarity semantic tokens and retain a condensed token sequence. Also, based on Token Merge, we obtained a source map $S$ for their spatial relationships between the raw token sequence $Z_L$ and the $Z_K$.

**Generating Mixing Mask with Source Matrix.** Since token merging aggregates non-similar tokens into compact representations $Z_K$, the resulting matrix preserves local feature structures more effectively. In contrast, the vanilla TopK selection adopts a greedy sampling strategy with linear complexity $\mathcal{O}(N)$, which discards low-ranked tokens directly and thus loses spatial relationships. Alternatively, the Bipartite Soft Matching (BSM) approach performs global pairwise matching with quadratic complexity, yielding a more balanced and globally optimal merging of tokens. To reconstruct the full-resolution attention map, we introduce a recovering function $\mathcal{R}_{K \to L}(\cdot, \cdot)$ that expands the merged attention map $A_K$ back to its original length $A_L$, which shows in Figure 5 and according to Eq. (6):

$$\hat{A}_L = \mathcal{R}_{K \to L}(A_K, S). \qquad (6)$$

Unlike discrete TopK sampling, our recovery mechanism propagates merged attention over the original token topology guided by similarity $S$, restoring richer spatial dependencies and contextual continuity, thus reducing information loss from hard selection. Based on the encoder with a token merge and attention recovery. We can generate the binary $\mathcal{M}$ according to Eq. (7):

$$\mathcal{M}_i = \begin{cases} 1, & \text{if } i \in \text{TopK}(\hat{A}_L, p), \\ 0, & \text{otherwise}, \end{cases} \qquad (7)$$

where $p$ denotes the selection number, $p = \lfloor \lambda * L \rfloor$, and $i$ denotes the index of sequence. Finally, we can mix the mini-batch and get the augmented data for $f_\theta(\cdot)$ training.

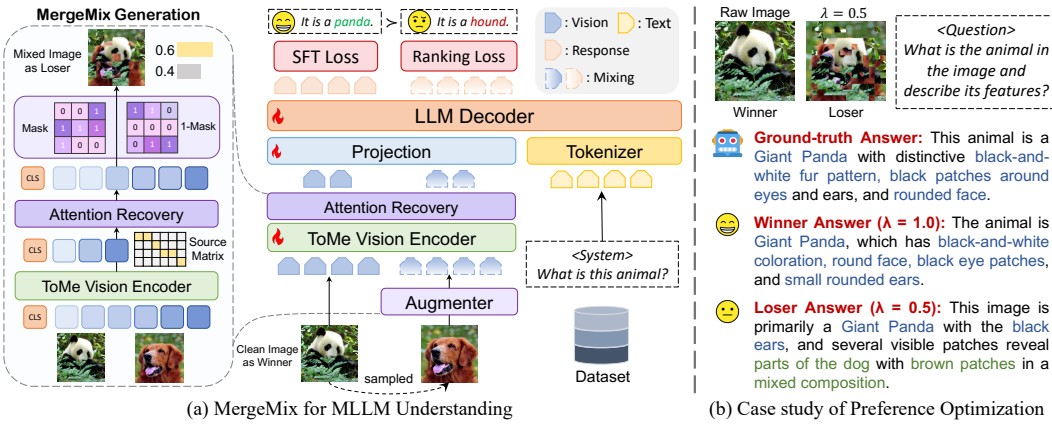

(a) MergeMix for MLLM Understanding       (b) Case study of Preference Optimization

Figure 3: **Overall illustration of MergeMix for MLLM. (a)** MergeMix performs attention-based mask mixing guided by the ToMe Vision Encoder, recovering token attention scores and generating a mixed image through an augmenter. Specifically, Token Merging hierarchically merges visual tokens via Bipartite Soft Matching (BSM) to enhance efficiency, which is trained with both the SFT and ranking losses. **(b)** Case study of preference data generated by MergeMix with LLaVA-v1.5-7B.

## 4.2 A Unified Augmentation Paradigm: From Image Classification to MLLMs

In this subsection, we describe the loss function $\mathcal{L}$. For the classification task and visual understanding, our final loss $\mathcal{L}_{\text{Total}}$ combines two losses: the main loss (one-hot cross entropy loss $\mathcal{L}_{\text{CE}}$ and $\mathcal{L}_{\text{SFT}}$) and the reformulated loss (mixup cross entropy loss $\mathcal{L}_{\text{MCE}}$ and ranking loss $\mathcal{L}_{\text{SimPO}}^{\text{Mix}}$). Figure 3 shows the pipeline of MergeMix for MLLM in detail.

**Re-scaling Policy for Mixing Ratio.** Under this optimization objective, the role of the mixing ratio $\lambda$ is to serve as a metric that quantifies the presence of feature information from the two samples. While this metric cannot directly reflect the true characteristics of the data, certain adaptive methods can constrain the model to generate mixed samples where the mixing ratio progressively approximates the target value (Jin et al., 2024a). Some works, like LUMix (Sun et al., 2022), DecoupleMix (Liu et al., 2023), and SUMix (Qin et al., 2024b), use a defined policy for some hand-crafted mixup methods, and find that it is more efficient than optimizing a better mask way.

Since we introduce a token-merging technology that inherently enables information aggregation and selection, the entire model training process must consider not only simple spatial ratios but also the degree of information integration within the model. So, we proposed a Gaussian-based sampling to refine the ratio, where the merged tokens and the mask values jointly control the `mean` and `std` as `mean`= $\frac{K}{L}$, and `std`= $\frac{p}{\sum_i^L \mathcal{M}}$. This smooth transition directly alleviates changes from linear mapping and yields more robust augmentations, with its formulation given as:

$$\hat{\lambda} \sim \mathcal{N}(\mu, \sigma), \quad \hat{\lambda} = \text{clip}\big(\frac{\hat{\lambda} - \min(\hat{\lambda})}{\max(\hat{\lambda}) - \min(\hat{\lambda}) + \tau}, \, 0, \, 1\big), \tag{8}$$

where the $\mathcal{N}(\cdot, \cdot)$ denotes a Gaussian function, $\mu$ and $\sigma$ represent the merged ratio and mixing ratio repetitively. $\tau$ as a hyperparameter, set to `1e-5`. Then, we obtain the re-scaled mixing ratio $\hat{\lambda}$ with spatial and ToMe model inherent features to optimize the model when training. In total, the loss of mixup training as:

$$\mathcal{L}_{\text{Total}} = \underbrace{\mathcal{L}_{\text{CE}}\big(f_\theta(\hat{x}), y_i\big) * \hat{\lambda} + \mathcal{L}_{\text{CE}}\big(f_\theta(\hat{x}), y_j\big) * (1 - \hat{\lambda})}_{\text{mce loss}} + \underbrace{\mathcal{L}_{\text{CE}}\big(f_\theta(x), y\big)}_{\text{one-hot loss}}. \tag{9}$$

**Aggregating Mixing Ratio within Preference Loss.** LLaVA (Liu et al., 2024b) uses a standard conditional language modeling loss for SFT. In MLLMs, we are given an instruction–response pair $(x, y)$, where $x$ denotes the multi-modal data, inducing vision and text, and $y = (y_1, y_{|y|})$ denotes the target response. The SFT loss is defined as:

$$\mathcal{L}_{\text{SFT}} = -\mathbb{E}_{(x,y)\sim\mathcal{D}}\left[\sum_{t=1}^{|y|} \log \pi_\theta\big(y_t \mid x, y_{<t}\big)\right]. \tag{10}$$

This objective needs to maximize the likelihood of GT responses, aiming to align the data. In Section 3, we introduced the DPO loss, which can be decomposed into two components: the SFT part and the ranking optimization part. In our approach, we replace the ranking component with SimPO (Meng et al., 2024), where $y$ denotes the target sequence (response) and $|y|$ denotes its length. Furthermore, since $\lambda$ reflects information similarity between augmented and raw image (interpreted as "loser degree" in MLLMs), we link it to $\gamma \to 1 - \hat{\lambda}$: larger $\lambda$ represent higher similarity and harder discrimination, reduces $\gamma$ to avoid over-optimization on trivial differences; smaller $\lambda$ represent greater dissimilarity and easier tasks increases $\gamma$ to strengthen constraints for clearer preference distinction. The mixed SimPO loss replacement is Eq. (11):

$$\mathcal{L}_{\text{SimPO}}^{\text{Mix}} = -\mathbb{E}_{(x,\hat{x},y)\sim\mathcal{D}}\left[\log \sigma\big(\frac{\beta}{|y|} \log \pi_\theta(y \mid x) - \frac{\beta}{|y|} \log \pi_\theta(y \mid \hat{x}) - (1 - \hat{\lambda})\big)\right]. \tag{11}$$

This reformulated loss strictness with sample difficulty, enabling more robust preference optimization. Finally, the total loss of our training paradigm is written as:

$$\mathcal{L}_{\text{Total}} = \mathcal{L}_{\text{SFT}} + \mathcal{L}_{\text{SimPO}}^{\text{Mix}}. \tag{12}$$

## 5 EXPERIMENTS

### 5.1 STATE-OF-THE-ART METHODS

**Image classification.** To evaluate the performance of MergeMix, we compared with some mainstream mixup methods, *i.e.* Mixup (Zhang et al., 2017), CutMix (Yun et al., 2019), FMix (Harris et al., 2020), SmoothMix (Lee et al., 2020), GridMix (Baek et al., 2021), ResizeMix (Qin et al., 2020), SaliencyMix (Uddin et al., 2021), Attentive-CutMix (Walawalkar et al., 2020), PuzzleMix (Kim et al., 2020), GuidedMixup (Kang & Kim, 2023), AutoMix (Liu et al., 2022) and AdAutoMix (Qin et al., 2024a). DeiT (Touvron et al., 2021), TransMix (Chen et al., 2022), SM-Mix (Chen et al., 2023), MixPro (Zhao et al., 2023) and TdAttenMix (Wang et al., 2025) for some ViT-based methods. The training configures about datasets and methods follows the open-source library OpenMixup (Li et al., 2022).

Table 1: Top-1 accuracy (%) of mixup methods training 200 epochs on CIFAR100 dataset with different model sizes, T/S/B/L denotes Tiny, Small, Base, and Large, respectively. The full results of training 600 epochs are in Table A1.

| Method | DeiT-T | DeiT-S | ViT-S | ViT-B | ViT-L |
|--------|--------|--------|-------|-------|-------|
| Vanilla | 64.70 | 65.81 | 62.64 | 63.33 | 61.83 |
| MixUp | 69.47 | 69.98 | 68.67 | 69.66 | 67.90 |
| CutMix | 75.98 | 74.21 | 69.67 | 72.18 | 68.97 |
| FMix | 72.73 | 70.41 | 68.41 | 68.62 | 66.12 |
| GridMix | 71.54 | 68.86 | 70.15 | 66.63 | 63.20 |
| ResizeMix | 69.42 | 68.54 | 67.86 | 63.72 | 63.48 |
| SaliencyMix | 69.83 | 69.78 | 70.14 | 68.75 | 67.12 |
| PuzzleMix | 73.40 | 73.60 | 70.92 | 71.13 | 69.77 |
| AutoMix | 72.91 | 76.24 | 68.44 | 73.40 | 72.10 |
| AdAutoMix | 72.83 | 72.63 | 69.66 | 71.43 | 69.69 |
| DeiT | 74.01 | 75.92 | 72.96 | 72.15 | 69.23 |
| TransMix | 75.31 | 76.17 | 74.15 | 72.87 | 71.40 |
| SMMix | 73.84 | 74.09 | 73.50 | 70.87 | 71.38 |
| MixPro | 74.78 | 75.26 | 73.49 | 73.18 | 72.28 |
| TdAttenMix | 73.63 | 73.32 | 73.11 | 72.19 | 72.12 |
| MergeMix | 77.46 | 78.68 | 77.02 | 75.75 | 76.19 |

Table 2: Top-1 accuracy (%) of mixup methods on the Stanford-Cars dataset. Full results of the CUB200 and FGVC-Aircrafts dataset in Table A2.

| Method | $\alpha$ | DeiT-S | ViT-B |
|--------|----------|--------|-------|
| Vanilla | − | 86.77 | 91.31 |
| MixUp | 1.0 | 87.73 | 91.36 |
| CutMix | 0.2 | 88.37 | 91.53 |
| SmoothMix | 0.2 | 86.39 | 90.88 |
| FMix | 0.2 | 87.18 | 91.36 |
| GridMix | 0.2 | 87.58 | 91.31 |
| ResizeMix | 1.0 | 87.45 | 91.59 |
| Attentive-CutMix | 2.0 | 87.35 | 90.29 |
| SaliencyMix | 0.2 | 87.94 | 91.47 |
| PuzzleMix | 1.0 | 88.60 | 91.83 |
| GuidedMix[ap] | 1.0 | 86.99 | 90.40 |
| DeiT | 0.2 | 88.72 | 92.17 |
| TransMix | 1.0 | 88.38 | 91.66 |
| SMMix | 1.0 | 88.76 | 91.93 |
| MixPro | 1.0 | 88.38 | 91.48 |
| TdAttenMix | 1.0 | 88.78 | 91.68 |
| MergeMix | 1.0 | 89.42 | 92.20 |

**MLLMs.** To evaluate the training paradigm that we proposed, we compare with three different system-level baselines: **(1)** SFT with different training paradigms on LLaVA, including LLaVA-NeXT-7/13B (Liu et al., 2024a), SeVa-7B (Zhu et al., 2024), SIMA (Wang et al., 2024b), and nSFT (Zhu et al., 2025b). **(2).** Token reduction on LLaVA, including LLaVA-PruMerge+ (Shang et al., 2024), VisionZip (Yang et al., 2025b), VisPrunner (Zhang et al., 2024b), VScan (Zhang et al., 2025a), and LLaVA-Mini (Zhang et al., 2025b). **(3).** RL training on Qwen2.5-VL-Instruction (Bai et al., 2025), including VisionThink (Yang et al., 2025c).

For all classification results, we report the top-1 test accuracy in the last 10 training epochs for each trial. To facilitate comparison, we mark the best and second best results in **bold** and cyan. For the LLaVA benchmark, we use the LLaVA official code, and for the Qwen2.5-VL-Instruction benchmark. We use lmms-eval (Zhang et al., 2024a) for evaluation.

## 5.2 DATASETS

In our paper, we mainly divided into 2 scenarios: Image Classification and MLLM Benchmark. The detailed information about datasets is described in Appendix B.1. For the image classification datasets, we choose 5 public classification datasets, including the small-scale dataset of CIFAR100 (Krizhevsky et al., 2009), the large-scale dataset of ImageNet-1K (Russakovsky et al., 2015), and the fine-grained datasets of CUB200 dataset (Wah et al., 2011), FGVC-Aircrafts dataset (Maji et al., 2013), and Stanford-Cars dataset (Krause et al., 2013). For the MLLM datasets, we choose 16 datasets, including visual question answering (VQAv2 (Goyal et al., 2017), GQA (Hudson & Manning, 2019), VizWiz (Gurari et al., 2018), ScienceVQA$^I$ (Lu et al., 2022), TextVQA (Singh et al., 2019), MME-RealWorldQA (Zhang et al., 2025c)), understanding (MME (Perception) (Yin et al., 2023), MMBench (Liu et al., 2025), MMBench$^{CN}$, MMBench$^{CC}$, POPE (F1 score) (Li et al., 2023b), SEED$^I$ (Li et al., 2023a), MM-Star (Chen et al., 2024b)), and reasoning (MMMU (Yue et al., 2024a), MMMU-Pro Standard (MMMU-Pro$^s$) (Yue et al., 2024b), MathVista (Lu et al., 2024)).

Table 3: The ImageNet-1K dataset classification results on Top1 Accuracy (Acc), Dynamic Forward, Throughput (TP/s) and FLOPs (G) in a NVIDIA A100. Forward$^{Dy.}$ denotes the metric through forward with dynamic.

| Method | DeiT-Small | | | |
|--------|------------|------|-----------|---------|
| | Forward$^{Dy.}$ | TP/s | FLOPs (G) | Acc (%) |
| Vanilla | ✗ | 1375.80 | 4.24 | 75.66 |
| MixUp | ✗ | 1374.54 | 4.24 | 77.80 |
| CutMix | ✗ | 1374.61 | 4.24 | 80.13 |
| DeiT | ✗ | 1374.20 | 4.24 | 79.80 |
| TransMix | ✗ | 1375.17 | 4.24 | 80.44 |
| SMMix | ✗ | 1373.93 | 4.24 | 79.36 |
| MixPro | ✗ | 1373.62 | 4.24 | 79.33 |
| MergeMix | ✓ | 1591.66 | 3.56 | 80.71 |

## 5.3 IMPLEMENTATIONS

In this subsection, we briefly introduce implementations on classification tasks and MLLM benchmarks (full details in Appendix B.2). For CIFAR100, ViT-based models (e.g., DeiT) use 224×224

Table 4: **Full system-level comparison results in LLaVA**. Compared with their counterparts. **AVG** denotes the average of the nine benchmarks for comprehensive comparison, except for MME, underline denotes MME with the sum of Perception and Cognition. Token$^i$ denotes training with the token number. Full results in Table A10, Table A11 and Table A12.

| Models | Token$^i$ | Image Question Answering | | | | | Benchmarks | | | | | AVG | Gain |
|---|---|---|---|---|---|---|---|---|---|---|---|---|---|
| | | VQAv2 | GQA | VizWiz | SciVQA$^I$ | TextVQA | MME | MMBench | MMBench$^{CN}$ | POPE | SEED$^I$ | | |
| **LLaVA Variants** | | | | | | | | | | | | | |
| LLaVA-7B | Full | 78.5 | 62.0 | 50.0 | 66.8 | 58.2 | 1510.7 | 64.3 | 58.3 | 85.87 | 66.19 | 65.57 | − |
| LLaVA-NeXT-7B | Full | 81.8 | 64.2 | 57.6 | 70.1 | 64.9 | 1519.0 | 67.4 | 60.6 | 86.5 | 70.2 | 69.3 | − |
| LLaVA-NeXT-13B | Full | 82.8 | 65.4 | 60.5 | 73.6 | 67.1 | 1575.0 | 70.0 | 64.4 | 86.2 | 71.9 | 71.3 | − |
| SeVa-7B | Full | − | 60.7 | − | 67.5 | 56.2 | 1450 | 65.6 | 59.2 | 86.7 | 65.8 | − | − |
| SIMA | Full | − | 62.2 | 54.4 | 68.1 | 58.3 | 1507.7 | 64.9 | 59.0 | 86.5 | 65.9 | − | − |
| nSFT | Full | − | 62.9 | − | 68.5 | 58.7 | 1531 | 67.1 | 61.0 | 86.8 | 66.2 | − | − |
| **LLaVA with Token Compressions** | | | | | | | | | | | | | |
| LLaVA-PruMerge+ | 144 | 76.8 | − | − | 68.3 | 57.1 | 1462.4 | 64.9 | − | 84.0 | − | − | − |
| VisionZip | 192 | 77.4 | 60.1 | − | 68.2 | 57.8 | 1834.0 | 63.4 | − | 84.9 | 57.1 | − | − |
| VisPrunner | 128 | 75.8 | 58.2 | 52.7 | 69.1 | 57.0 | 1461.4 | 62.7 | 57.3 | 84.6 | − | − | − |
| VScan | 192 | 77.8 | 60.6 | 50.4 | 68.6 | 57.7 | 1806.0 | 63.9 | 57.4 | 86.2 | − | − | − |
| LLaVA-Mini | 1 | 77.6 | 60.9 | 56.2 | 70.4 | 57.0 | 1466.0 | 65.6 | − | 84.4 | 58.5 | − | − |
| **LLaVA with Augmentations & Ranking Loss** | | | | | | | | | | | | | |
| SFT Vision | Full | **79.32** | **62.98** | 47.45 | 70.05 | 57.17 | **1490.88** | 66.26 | 60.05 | 86.18 | 67.32 | 66.31 | **+0.74** |
| + MixUp | Full | 79.27 | 62.58 | 44.95 | 69.41 | 57.39 | 1483.20 | 65.72 | 58.24 | 86.27 | 66.73 | 65.62 | **+0.05** |
| + CutMix | Full | 79.18 | 62.40 | 45.04 | **70.60** | 57.06 | 1452.31 | 66.32 | 58.24 | **86.47** | 67.22 | 65.84 | **+0.27** |
| + ResizeMix | Full | 77.78 | 61.66 | 44.43 | 68.91 | 55.11 | 1436.09 | 63.91 | 55.41 | 86.01 | 63.91 | 64.13 | **-1.44** |
| + MergeMix | Full | 79.24 | 62.44 | 47.69 | 69.86 | **57.56** | 1479.97 | **66.58** | **60.65** | 86.10 | **67.47** | **66.40** | **+0.83** |
| SFT Vision | 288 | 78.6 | **62.47** | 48.15 | 69.51 | 56.41 | **1486.24** | 66.32 | 57.98 | **87.37** | 66.75 | 65.95 | **+0.38** |
| + MixUp | 288 | 78.51 | 62.07 | 51.1 | 68.47 | 56.54 | 1459.06 | 65.63 | **59.53** | 86.86 | 66.06 | 66.08 | **+0.51** |
| + CutMix | 288 | 78.58 | 62.39 | 50.53 | 70.2 | 55.95 | 1414.72 | **66.92** | 59.53 | 86.56 | 66.2 | 66.31 | **+0.74** |
| + ResizeMix | 288 | 76.39 | 61.05 | 45.48 | 68.07 | 54.60 | 1447.35 | 63.31 | 51.97 | 86.57 | 62.54 | 63.33 | **-2.24** |
| + MergeMix | 288 | **78.61** | 62.18 | **52.14** | 69.61 | **56.85** | 1453.97 | **66.58** | 59.02 | 86.47 | 66.63 | **66.45** | **+0.88** |

Table 5: **Full system-level comparison results in Qwen2.5-VL-Instruction (Qwen2.5-VL-Ins)**. **AVG** denotes the average of the nine benchmarks for comprehensive comparison.

| Models | MMStar | MMBench | MMBench$^{CN}$ | MMBench$^{CC}$ | POPE | RWQA | MMMU | MMMU-Pro$^s$ | MathVista | AVG | Gain |
|---|---|---|---|---|---|---|---|---|---|---|---|
| Qwen2.5-VL-Ins-7B | 62.42 | **84.02** | 80.41 | 62.94 | 86.38 | 68.63 | 50.3 | 36.42 | 19.2 | 61.19 | − |
| VisionThink-7B | 61.00 | 82.73 | **81.01** | 64.5 | **87.65** | **69.28** | 51.0 | **37.27** | 23.8 | 62.03 | **+0.84** |
| SFT Vision | 62.66 | 83.41 | **81.01** | 63.52 | 87.69 | 68.63 | 50.89 | 36.7 | **38.4** | 63.66 | **+1.47** |
| + MergeMix | **62.92** | **84.19** | **81.18** | **64.31** | 87.28 | **70.46** | 51.0 | 37.46 | 37.8 | **64.07** | **+2.88** |

images, trained with AdamW (weight decay 0.05), batch size 100, 200/600 epochs, Random-Flip/RandomCrop/RandAugment (Cubuk et al., 2020), and learning rates: 1e-3 (DeiT-Tiny/Small, cosine), 5e-4 (ViT-Small/Base), 2e-4 (ViT-Large). For ImageNet-1K, settings are consistent except 1e-3 learning rate, batch size 1024, 300 epochs (DeiT-Tiny/Small). For fine-grained datasets (CUB200, FGVC-Aircrafts, Stanford-Cars), DeiT-Small/ViT-Base are fine-tuned 200 epochs (batch size 16, 1e-5 lr) with PyTorch pretrained weights (Paszke et al., 2019).

Following LLaVA-v1.5, we use Vicuna-v1.5 7B (Chiang et al., 2023) (language decoder) and a 2-layer pre-trained MLP (visual-textual alignment, 1 epoch on `LCS-558K`), with a pre-trained CLIP vision encoder. SFT uses 1 epoch on `llava-v1.5-mix665k` (2e-5 lr, batch size 64, AdamW, 0.03 warmup, cosine scheduler), unfreezing the vision encoder (unlike LLaVA). For Qwen2.5-VL-Instruction, official checkpoints are fine-tuned 0.1 epoch on `llava-v1.5-mix665k` (similar settings); learning rates: 2e-6 (vision encoder), 1e-5 (LLM decoder/merger).

## 5.4 RESULTS OF IMAGE CLASSIFICATION

We did the experiments on three classification datasets on a small-scale dataset (CIFAR100), a large-scale dataset (ImageNet-1K), and a fine-grained dataset (Stanford-Cars). **(1) CIFAR100:** Table 1 and Table A1 shows the part and full classification results respectively. MergeMix brings the **+2.15%**, **+2.51%** gains compared with TranMix on DeiT models. Gains **+2.87%**, **+2.88%** and

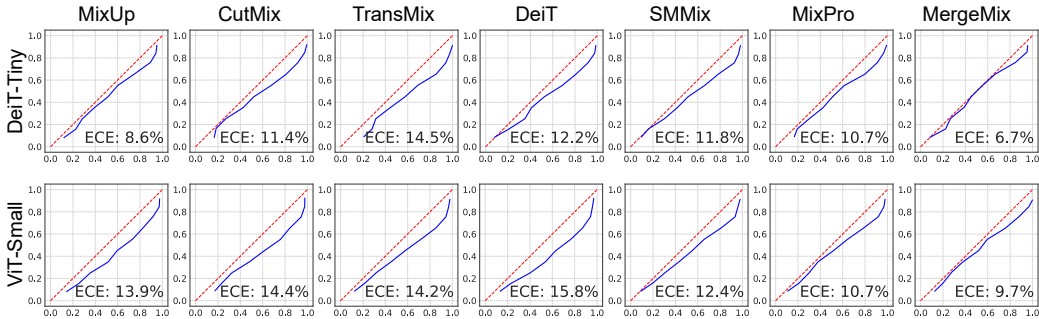

Figure 4: The confidence plots of mixup variants and MergeMix on the CIFAR100 dataset using DeiT-Tiny and ViT-Small. The red line indicates the expected prediction tendency.

Table 6: **The calibration results of LLaVA-v1.5-7B** on POPE, ScienceVQA$^I$, GQA & SEED$^I$. rl denotes training with ranking loss.

| Method | GQA | POPE | SEED$^I$ | ScienceVQA$^I$ |
|---|---|---|---|---|
| Baseline | 14.57 | 13.16 | 33.79 | 28.09 |
| **Training with Full Vision Tokens** | | | | |
| SFT Vision | 8.52 | 12.82 | 32.67 | 21.66 |
| +MixUp$^{rl}$ | **6.09** | 12.72 | 33.26 | **21.51** |
| +CutMix$^{rl}$ | 6.74 | **12.62** | 32.77 | 24.71 |
| +ResizeMix$^{rl}$ | 12.53 | 13.17 | 36.08 | 24.58 |
| +MergeMix$^{rl}$ | 6.50 | 12.91 | **32.52** | 23.66 |
| **Training with 50% Vision Tokens** | | | | |
| SFT Vision | 18.13 | **12.67** | 34.41 | 24.28 |
| +MixUp$^{rl}$ | 13.40 | 12.74 | 33.60 | 22.61 |
| +CutMix$^{rl}$ | 10.48 | 12.67 | 33.83 | **20.63** |
| +ResizeMix$^{rl}$ | 12.60 | 12.97 | 37.41 | 23.74 |
| +MergeMix$^{rl}$ | **10.34** | 12.76 | **33.37** | 25.22 |

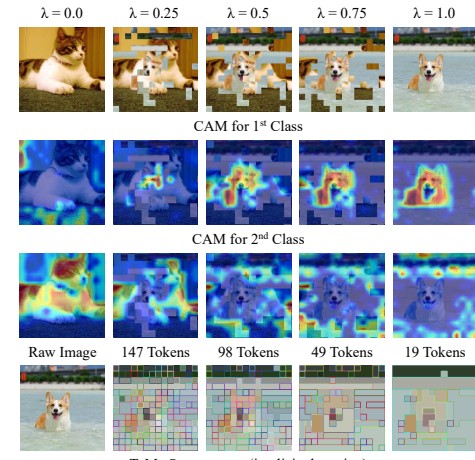

Figure 5: Visualization of MergeMix on different mixing ratios, including mixed images, Grad-CAM of top-2 logits, and ToMe source maps.

+4.79% on ViT models. All the results in Table 1 are trained for 200 epochs on the CIFAR100 dataset. **(2) Stanford-Cars:** Table 2 shows the fine-grain classification results on Stanford-Cars dataset. MergeMix achieves the **88.42%** and **92.20%** accuracy compared with other mixup methods. About the results of the CUB200 dataset and the FGVC-Aircrafts dataset in Table A2. **(3) ImageNet-1k:** Table 3 shows the results of accuracy, throughput, and flops on the ImageNet-1K dataset. It is notable that MergeMix brings a **+0.27%** gains and reduces **-0.68**G Flops compared with TransMix, and can also see other mixups with less throughput since they bring extra cost, but MergeMix has a high throughput of 1591.66 TP/s.

## 5.5 RESULTS OF MLLM BENCHMARKS

We chose two mainstream MLLMs for our experiments on the VQA tasks and reasoning. Table 4 shows the results of LLaVA benchmarks, with different vision tokens for training. With the setting of full vision tokens on the training stage, our method achieves an average gain of **+0.83%**. When reducing the vision token to 288, our method still performs well compared with SFT. A full comparison of results in Table A10, Table A11, and Table A12. In those different settings, MergeMix achieves an average performance of **66.84%**, improving over the LLaVA baseline by **+1.27%**. Table 5 shows the results of some VQA tasks and reasoning with Qwen2.5-VL-Instruction. MergeMix achieves an average gain of **+2.88%** over Qwen2.5-VL-Instruction. For the MathVista reasoning task, we reported the results without `LLM-as-the-judge-eval`. For the MMMU and MMMU-Pro tasks, we can achieve results on par with methods targeting reasoning improvements.

## 5.6 RESULTS OF CALIBRATION

DNNs are prone to overconfidence in classification tasks. Mainfold Mixup (Verma et al., 2019) found that the mixup methods can effectively alleviate this problem. To this end, we compute the

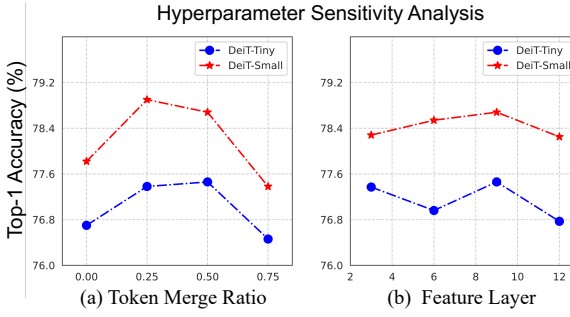

Figure 6: Sensitivity analysis study of 2 hyperparameters of MergMix. *Left:* Different merge ratios of backbones. *Right:* Attention score obtained from feature layer. Those results are from training for 200 epochs.

Table 7: Ablation study of MergeMix on classification trained 200 epochs.

| Module | DeiT-Small | ViT-Small |
|---|---|---|
| Vanilla | 65.81 | 62.64 |
| + TopK | 75.80 | 75.19 |
| + ToMe | 76.45 | 76.46 |
| + Re-Scaling $\lambda$ | **78.68** | **77.02** |

Table 8: Ablation study of MLLM training paradigm on LLaVA benchmark.

| Method | VizWiz | SciVQA$^I$ | MMBench |
|---|---|---|---|
| LLaVA-v1.5-7B | 50.0 | 66.8 | 64.3 |
| + ToMe | 50.45 | 68.86 | 62.8 |
| + SFT | 48.15 | 69.51 | 66.32 |
| + MergeMix$^{rl}$ | **52.14** | **69.61** | **66.58** |

Expected Calibration Error (ECE (Guo et al., 2017)) of various mixup approaches on the CIFAR100 dataset for image classification. Also, to further analyze the calibration of MLLMs, we implement four short answer tasks, POPE, GQA, ScienceVQA, and SEED. Figure 4 shows the results of DeiT-Tiny and ViT-Small models trained for 200 epochs, showing that MergeMix obtains the best calibration of **6.7**% and **9.7**% in those ViT-specific mixup methods. (*i.e.*, TransMix, SMMix, and MixPro). Table 6 shows the results of the LLaVA baseline, LLaVA with SFT, and LLaVA with our approach. SFT reduces the ECE when tuning the vision encoder, with augmentation and ranking loss, which can be better since we bring in the reward signal for the model. The more comprehensive results of CIFAR100 and MLLM benchmark we plot in Table A3 and Table A9.

## 5.7 ABLATION STUDY

The ablation study mainly focuses on three things. **(a)** Token merge module and optimized mixing ratio, whether efficient for image classification task; **(b)** Exploring the ability of vision encoder and the proposed training paradigm. For the image classification scenario, Table 7 shows that compared with TopK sampling, our token merge can improve performance with **+0.55**%, **+1.27**% gains respectively, which means token merge smooths the discrete attention score. The re-scaling mixing ratio further gains **+2.23**%, **+0.56**% on the CIFAR100 dataset. For the paradigm, we validate the token merge for the LLaVA-v1.5 7B model, further explore the training with an unfrozen vision encoder, and the ranking loss. Table 8 shows that, compared with vanilla Token Merge, unfreezing the vision encoder can perform better than freezing. The augmentation and ranking loss bring more performance than only the SFT loss; **(c)** For further exploring the performance of hyperparameters. We also evaluated the sensitivity of hyperparameters on MergeMix, *i.e.*, merged tokens and feature layer for better performance. Figure 6 shows the results of those hyperparameters.

## 6 CONCLUSION

This paper presents *MergeMix*, a unified augmentation for both image classification and MLLM alignment with token merge, also bridging the SFT and RL by building the preference pairs. Optimizing models through the mixed image and the raw image via a ranking loss. Extensive experiments demonstrate that MergeMix not only improves the performance on classic image classification tasks but also achieves a beneficial alignment and generalization on MLLM benchmarks. MergeMix provides a promising step toward a scalable, robust training paradigm for the multi-modal system.

**Future Works** There remain limitations in MergeMix for MLLMs. In future work, we will address them from two directions: **(1)** Data level: MergeMix currently enhances only the image modality, while text remains untouched. Extending mixup to the text modality could provide more fine-grained optimization. **(2)** Model level: The token-merging policy is static and unlearned. Making it learnable may further improve the mixing capability.

ACKNOWLEDGEMENT

This paper is supported by Young Scientists Fund of the National Natural Science Foundation of China (NSFC) (No. 62506305), Zhejiang Leading Innovative and Entrepreneur Team Introduction Program (No. 2024R01007), Key Research and Development Program of Zhejiang Province (No. 2025C01026), Scientific Research Project of Westlake University (No. WU2025WF003), Chinese Association for Artificial Intelligence (CAAI) & Ant Group Research Fund - AGI Track (No. 2025CAAI-ANT-13). It is also supported by the research funds of the National Talent Program and Hangzhou Municipal Talent Program.

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

## A  DECLARATION OF LLM USAGE

We use the Large Language Models (LLMs) for this paper to serve one purpose: to aid and polish the paper writing. We use the LLMs in a very limited capacity, restricted to minor editing of grammar, phrasing, and readability. We do not involve the LLMs in designing the method, developing theoretical results, and conducting experiments.

## B  DETAIL INFORMATION

### B.1  DATASETS

**Image Classification.** We choose five mainstream classification datasets: **(i)** CIFAR100 dataset (Krizhevsky et al., 2009) consists of 100 classes of color images with a resolution of $32 \times 32$ pixels, containing 50,000 training images and 10,000 test images. **(ii)** ImageNet-1k dataset (Russakovsky et al., 2015) consists of 1,000 classes with varied image resolutions commonly cropped to $224 \times 224$ pixels, containing 1,281,167 training images and 50,000 test images. **(iii)** CUB200 dataset (Wah et al., 2011) contains 200 bird species, including total 11,788 images, we divided 5,994 images as training images and 5,794 images as test images, **(iv)** FGVC-Aircrafts dataset (Maji et al., 2013) contains 100 aircraft model classes, including 6,667 training images and 3,333 test images, **(v)** Stanford-Cars dataset (Krause et al., 2013) contains 196 car model classes, including 8,144 training images and 8,041 test images. All the fine-grained datasets we used set the resolution as $224 \times 224$ pixels for training and testing.

**MLLM Benchmark.** we conducted various experiments on LLaVA Benchmark and lmms-eval (Zhang et al., 2024a), which based on 16 datasets: **(i)** VQAv2 dataset (Goyal et al., 2017) contains 204,721 training images, 22,000 validation images, and 40,504 test images, **(ii)** GQA dataset (Hudson & Manning, 2019) focuses on graph-based reasoning with 220,000 training images and 150,000 validation/test questions, **(iii)** VizWiz dataset (Gurari et al., 2018) images are captured by mobile devices with questions from visually impaired users (31,173 training images), **(iv)** ScienceVQA dataset (Lu et al., 2022), **(v)** TextVQA dataset (Singh et al., 2019), and **(vi)** SEED dataset (Li et al., 2023a) contain 21,208, 28,408, and 15,000 training images, respectively, emphasizing scientific reasoning, text understanding, and multi-modal reasoning. **(vii)** MME dataset (Yin et al., 2023) and **(viii)** MMBench dataset (Liu et al., 2025) provide general multi-modal evaluation, **(ix)** MMBench$^{CN}$ dataset as the Chinese version of MMbench, **(x)** MMBench$^{CC}$ dataset as the Cross Check version of MMbench. **(xi)** POPE dataset (Li et al., 2023b) evaluates performance on prompt-driven tasks with zero-shot and few-shot settings. **(xii) & (xiii)** MMMU & MMMU-Pro datasets (Yue et al., 2024a;b) are a multi-modal reasoning benchmark with college-level exam questions, **(xiv)** MME-RealWorldQA dataset (Zhang et al., 2025c) emphasizes real-world, long-tail visual understanding in everyday scenarios. **(xv)** MMStar dataset (Chen et al., 2024b) uses testing star-level multi-modal reasoning across diverse tasks. **(xvi)** MathVista dataset (Lu et al., 2024) for the visual mathematical reasoning by involving geometry, algebra, and charts. For the LLaVA benchmark, all images are typically cropped or resized to $336 \times 336$ pixels for training and evaluation since the CLIP (Radford et al., 2021) is the vision encoder. For the Qwen2.5-VL-Instruction benchmark, the images are dynamically scaled by the Qwen-VL model.

### B.2  IMPLEMENTATIONS

**Classification tasks:** **(i)** For the CIFAR100 dataset, aiming to be suitable for training ViT-based approaches, *e.g.*, DeiT, we resize images to $224 \times 224$ and train them with the AdamW optimizer with weight decay of 0.05, batch size of 100, and total training of 200 epochs and 600 epochs. Uses RandomFlip and RandomCrop as basic augmentations, and additionally, we use RandAugment (Cubuk et al., 2020). For DeiT-Tiny and DeiT-Small, we use the learning rate of 1e-3 with a dynamic cosine scheduler. For ViT-Small and ViT-Base models, we set the learning rate to 5e-4, the learning rate of ViT-Large up to 2e-4, all dynamically adjusted by a cosine scheduler. **(ii)** For the ImageNet-1K dataset, the dataset settings are the same as CIFAR100, but we use the learning rate as 1e-3, batch size of 1024, and a total training of 300 epochs for DeiT-Tiny and DeiT-Small with AdamW optimizer with weight decay of 0.05. **(iii)** For all fine-grain datasets, *i.e.*, CUB-200 dataset, FGVC-Aircrafts dataset, and Stanford-Cars dataset, we fine-tune the DeiT-Small and ViT-

Table A1: Top-1 accuracy (%) of mixup methods on CIFAR-100 dataset under DeiT-Tiny/Small, ViT-Small/Base/Large different model sizes. The $\alpha$ parameter of the Beta distribution follows the setting in OpenMixup (Li et al., 2022) setting.

| Method | DeiT-Tiny | | DeiT-Small | | ViT-Small | | ViT-Base | | ViT-Large |
| | 200 epochs | 600 epochs | 200 epochs | 600 epochs | 200 epochs | 600 epochs | 200 epochs | 600 epochs | 200 epochs |
|---|---|---|---|---|---|---|---|---|---|
| Vanilla | 64.70 | 66.70 | 65.81 | 68.50 | 62.64 | 66.32 | 63.33 | 66.47 | 61.83 |
| MixUp | 69.47 | 73.06 | 69.98 | 76.35 | 68.67 | 73.57 | 69.66 | 73.90 | 67.90 |
| CutMix | 75.98 | 79.60 | 74.21 | 79.54 | 69.67 | 76.66 | 72.18 | 71.94 | 68.97 |
| FMix | 72.73 | 77.24 | 70.41 | 74.31 | 68.41 | 72.55 | 68.62 | 71.10 | 66.12 |
| GridMix | 71.54 | 76.23 | 68.86 | 74.96 | 70.15 | 68.23 | 66.63 | 68.49 | 63.20 |
| ResizeMix | 69.42 | 72.98 | 68.54 | 71.95 | 67.86 | 69.09 | 63.72 | 69.33 | 63.48 |
| SaliencyMix | 69.83 | 75.45 | 69.78 | 76.60 | 70.14 | 74.09 | 68.75 | 75.50 | 67.12 |
| PuzzleMix | 73.40 | 79.96 | 73.60 | **81.01** | 70.92 | 78.44 | 71.13 | 79.49 | 69.77 |
| AutoMix | 72.91 | 81.16 | 76.24 | 80.91 | 68.44 | 77.73 | 73.40 | – | 72.10 |
| AdAutoMix | 72.83 | 77.97 | 72.63 | 78.94 | 69.66 | – | 71.43 | – | 69.69 |
| DeiT | 74.01 | 79.90 | 75.92 | 79.54 | 72.96 | 77.60 | 72.15 | 76.26 | 69.23 |
| TransMix | 75.31 | 80.66 | 76.17 | 79.33 | 74.15 | 78.27 | 72.87 | 77.89 | 71.40 |
| SMMix | 73.84 | 78.62 | 74.09 | 79.84 | 73.50 | 79.65 | 70.87 | 78.18 | 71.38 |
| MixPro | 74.78 | 80.19 | 75.26 | 79.55 | 73.49 | 80.02 | 73.18 | 78.69 | 72.28 |
| MergeMix | **77.46** | **81.20** | **78.68** | 80.39 | **77.02** | **81.44** | **75.75** | **79.59** | **76.19** |

Table A2: Top-1 accuracy (%) of mixup methods on Fine-Grained datasets: CUB200, FGVC-Aircrafts, and Stanford-Cars.

| Method | $\alpha$ | CUB200 | | FGVC-Aircrafts | | Stanford-Cars | |
| | | DeiT-Small | ViT-Base | DeiT-Small | ViT-Base | DeiT-Small | ViT-Base |
|---|---|---|---|---|---|---|---|
| Vanilla | – | 82.05 | 88.00 | 77.59 | 80.86 | 86.77 | 91.31 |
| MixUp | 1.0 | 84.31 | **88.75** | 78.52 | **82.18** | 87.73 | 91.36 |
| CutMix | 0.2 | 81.69 | 87.76 | 75.67 | 80.08 | 88.37 | 91.53 |
| SmoothMix | 0.2 | 83.87 | 87.02 | 75.31 | 76.72 | 86.39 | 90.88 |
| FMix | 0.2 | 82.64 | 88.68 | 77.08 | 79.33 | 87.18 | 91.36 |
| GridMix | 0.2 | 82.34 | 87.23 | 75.85 | 78.49 | 87.58 | 91.31 |
| ResizeMix | 1.0 | 82.15 | 87.61 | 74.59 | 77.62 | 87.45 | 91.59 |
| Attentive-CutMix | 2.0 | 82.83 | 87.47 | 75.04 | 76.06 | 87.35 | 90.29 |
| SaliencyMix | 0.2 | 82.34 | 87.92 | 77.98 | 79.81 | 87.94 | 91.47 |
| PuzzleMix | 1.0 | 84.39 | 88.23 | 78.28 | 81.27 | 88.60 | 91.83 |
| GuidedMix[ap] | 1.0 | 84.71 | 88.26 | 77.05 | 79.24 | 86.99 | 90.40 |
| DeiT | 0.2 | 84.04 | 88.47 | 75.89 | 81.07 | 88.72 | 92.17 |
| TransMix | 1.0 | 83.34 | 88.10 | 75.73 | 77.77 | 88.38 | 91.66 |
| SMMix | 1.0 | 82.88 | 88.35 | 76.42 | 78.40 | 88.76 | 91.93 |
| MixPro | 1.0 | 82.31 | 86.93 | 75.25 | 75.97 | 88.38 | 91.48 |
| MergeMix | 1.0 | **85.40** | 88.40 | **80.92** | 81.97 | **89.42** | **92.20** |

Base model for 200 epochs with a batch size of 16, learning rate of 1e-5, loading the pre-trained model weight from PyTorch (Paszke et al., 2019).

**MLLM benchmark:** Following the LLaVA-v1.5 settings, we use a pre-trained Vicuna-v1.5 7B (Chiang et al., 2023) as the language decoder, which uses a pre-trained $2 \times$ MLP as the projection for aligning the vision and text modistes, which was trained for one epoch on `LCS-558K`. For the vision encoder, we use a pre-trained CLIP encoder and extract the visual representation from the input images. For SFT, the learning rate was set as 2e-5, the batch size was 64, and training one epoch on `llava-v1.5-mix665k` dataset, uses AdamW optimizer with (0.9, 0.999) betas and epsilon of 1e-8, warmup ratio of 0.03 with a cosine scheduler. The difference from LLaVA is that we unfreeze the vision encoder during training. About Qwen2.5-VL-Instruction, we fine-tune with `llava-v1.5-mix665k` dataset for 0.1 epoch, uses AdamW optimizer with (0.9, 0.999) betas and epsilon of 1e-8 like LLaVA, warmup with 0.03 ratio. The learning rate of the vision encoder, LLM decoder, and merger were set to 2e-6, 1e-5, and 1e-5, respectively.

**Algorithm 1** MergeMix for Image Classification

```
# Inputs:  vision model fθ(·),
training parameters θ of model,
dataset D, mixup parameter α;
output:  updated fθ(·)
#   sample a paired mini-batch from
D (two images and labels)
1: for (xi,yi), (xj,yj) in
DataLoader(D):
#   sample mixup ratio λ (shared
with Alg. 2)
2:    λ ∼ Beta(α,α)
3:    M,λ̂ = P(xi,xj,λ)
#   lines 3--4:  MergeMix
augmentation (same core as Alg. 2)
4:    x̂ = M ⊙ x_i + (1 − M) ⊙ x_j
5:    logits = fθ(x̂)
#   main supervised loss on mixed
sample (parallels L_SFT in Alg. 2)
6:    L_MCE  =  λ̂ · L_CE(logits, yi) + (1 − λ̂) ·
L_CE(logits, yj)
#   optional regularizer for
augmentation consistency
7:    L_CE = fθ(x,y)
8:    L_Total = L_MCE + L_CE
9:    L_Total.backward()
10:  optimizer.step()
11:  optimizer.zero_grad()
```

**Algorithm 2** MergeMix for MLLM Alignment

```
# Inputs:  MLLM πθ(·), learnable
parameters θ of model, dataset D of
(x,q,T); output:  updated πθ(·)
#   iterate over triples
1: for (x,q,T) in DataLoader(D):
#   sample an auxiliary image xi
(mirrors mix partner in Alg. 1)
2:    (xi,_) = randomSample(D)
3:    M = P(x,xi)
#   lines 3--4:  same MergeMix
augmentation core as Alg. 1
4:    x̂ = M ⊙ x + (1 − M) ⊙ xi
#   winner (raw) vs. loser (mixed)
outputs for the same prompt
5:    Yw = πθ(x,q)
6:    Yl = πθ(x̂,q)
#   supervised fine-tuning loss
(analogous to L_CE in Alg. 1)
7:    L_SFT = L_CE(Yw,T)
8:    sw = AvgLogProb(Yw,T)
9:    sl = AvgLogProb(Yl,T)
#   preference loss (SimPO) that
prefers winner over loser
10:   L_SimPO^Mix = − log σ(sw − sl − λ)
11:   L_Total = L_SFT + L_SimPO^Mix
12:   L_Total.backward()
13:  optimizer.step()
14:  optimizer.zero_grad()
```

### B.3 Algorithms

Algorithm 1 and Algorithm 2 show the pseudo codes for MergeMix on both image classification and preference tuning.

### B.4 Future work

There are still some shortcomings in MergeMix for MLLMs. In future work, we will explore this from: **(1)** From the data perspective: MergeMix focuses on the enhancement of the image modality during training, while text inputs still remain raw. How to extend mixup to the text modality in MLLM tasks needs to be solved, as this can provide more fine-grained optimization guidelines. **(2)** From the model perspective: The token merging is still static and unlearned. Improving the merging strategy to make it learnable via metrics or backpropagation could enhance the token merging ability for mixing. **(3)** How to use sampling to find the optimal candidate as the loser from multiple augmented data, like RL-based methods (Feng et al., 2025b;a) and how to construct high-quality losers when adopting image editing methods (Wang et al., 2026) are also promising ways to obtain the preference pairs.

## C Image Classification

### C.1 Extensive Results

Table A1 shows the full results of 200 epochs and 600 epochs training on the CIFAR100 dataset using different ViT models. Table A2 shows the three fine-grain datasets, *i.e.*, CUB200, FGVC-Aircrafts, and Stanford-Cars. It is easily found that MergeMix can achieve the SOTA on

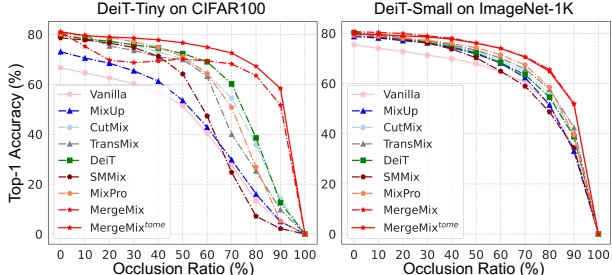

Figure A1: Robustness against image occlusion classification results with different occlusion ratios for different mixup methods based on DeiT-Tiny (left) and DeiT-Small (right) on CIFAR100 and ImageNet-1K datasets.

Table A3: The calibration results of ViT-based mixup methods on CIFAR-100 & ImageNet-1K, with training 200 and 300 epochs respectively. tome denotes inference with token merging.

| Models | Dataset | Epochs | MixUp | CutMix | TransMix | DeiT | SMMix | MixPro | MergeMix | MergeMix[tome] |
|--------|---------|--------|-------|--------|----------|------|-------|--------|----------|-------------|
| DeiT-Tiny | CIFAR100 | 200 | 8.64 | 11.42 | 14.52 | 12.24 | 11.80 | 10.68 | **7.08** | **6.73** |
| ViT-Small | CIFAR100 | 200 | 13.89 | 14.43 | 14.22 | 15.76 | 12.41 | 10.65 | **9.69** | **10.14** |
| ViT-Large | CIFAR100 | 200 | 4.76 | 7.36 | 16.44 | 6.71 | 7.70 | 6.22 | **4.65** | **4.70** |
| DeiT-Tiny | CIFAR100 | 600 | 6.44 | 6.01 | 7.56 | 8.53 | 8.35 | 6.38 | **5.45** | **5.46** |
| ViT-Small | CIFAR100 | 600 | 9.01 | 6.45 | 9.12 | 10.22 | 9.55 | 7.42 | **5.50** | **6.04** |
| DeiT-Small | ImageNet-1K | 300 | 5.66 | **4.19** | 7.57 | 8.24 | 6.53 | 5.17 | 6.00 | **4.52** |

Table A4: The Top-1 accuracy of DeiT-Tiny trained by various Mixup approaches on the CIFAR100 dataset with different occlusion ratios. tome denotes inference with token merging.

| Method | DeiT-Tiny Trained 200 epochs | | | | | | | | | |
|--------|------|------|------|------|------|------|------|------|------|------|
| | 0% | 10% | 20% | 30% | 40% | 50% | 60% | 70% | 80% | 90% |
| Vanilla | 66.68 | 64.54 | 62.57 | 60.20 | 56.96 | 51.41 | 40.32 | 27.75 | 13.25 | 4.99 |
| MixUp | 73.06 | 70.52 | 68.45 | 65.39 | 61.27 | 53.55 | 42.80 | 29.77 | 16.01 | 5.01 |
| CutMix | 79.58 | 78.64 | 77.10 | 75.83 | 72.96 | 70.25 | 64.40 | 54.39 | 35.82 | 14.08 |
| FMix | 77.14 | 76.01 | 74.62 | 73.33 | 71.27 | 67.72 | 63.17 | 56.18 | 42.12 | 17.32 |
| GridMix | 76.13 | 73.79 | 71.94 | 69.36 | 66.36 | 62.34 | 56.02 | 47.30 | 32.52 | 14.35 |
| ResizeMix | 72.93 | 71.82 | 70.53 | 69.67 | 67.77 | 65.22 | 59.87 | 50.26 | 31.26 | 9.72 |
| SaliencyMix | 75.41 | 74.63 | 73.57 | 72.14 | 69.47 | 65.05 | 58.08 | 44.82 | 24.14 | 7.03 |
| Attentive-CutMix | 80.27 | 79.13 | 77.94 | 76.98 | 75.45 | 71.01 | 57.75 | 33.10 | 12.03 | 4.02 |
| PuzzleMix | 79.97 | 78.18 | 77.36 | 76.04 | 73.74 | 70.74 | 65.83 | 56.75 | 40.43 | 19.41 |
| AutoMix | **81.12** | 78.37 | 78.40 | **78.16** | **77.65** | **77.09** | **74.63** | **71.64** | **67.52** | **55.61** |
| TransMix | 80.66 | **79.77** | 75.33 | 73.60 | 70.98 | 69.95 | 63.20 | 40.02 | 25.28 | 9.77 |
| DeiT | 79.80 | 78.20 | 77.29 | 75.83 | 74.59 | 72.31 | 69.05 | 60.18 | 38.55 | 12.53 |
| SMMix | 78.62 | 77.78 | 76.51 | 74.78 | 71.25 | 64.17 | 47.28 | 24.72 | 7.12 | 2.19 |
| MixPro | 80.14 | **79.61** | **78.78** | 77.06 | 74.81 | 71.18 | 64.25 | 50.75 | 26.80 | 5.19 |
| MergeMix | 80.88 | 75.19 | 69.71 | 68.72 | 69.33 | 70.17 | 69.36 | 68.20 | 63.49 | 51.72 |
| MergeMix[tome] | **81.12** | 79.49 | **78.96** | **78.57** | **77.81** | **76.65** | **74.89** | **72.58** | **67.24** | **58.32** |

lots of models. Also, the speed of overfitting is a significant improvement over other methods, which means the token merge can gather useful information and reduce some redundant tokens.

Table A5: The Top-1 accuracy of ViT-Small trained by various Mixup approaches on the CIFAR100 dataset with different occlusion ratios. tome denotes inference with token merging.

| Method | ViT-Samll Trained 200 epochs | | | | | | | | | |
|--------|------|------|------|------|------|------|------|------|------|------|
| | 0% | 10% | 20% | 30% | 40% | 50% | 60% | 70% | 80% | 90% |
| Vanilla | 66.24 | 62.51 | 60.37 | 58.59 | 56.68 | 53.61 | 49.70 | 39.70 | 22.60 | 7.55 |
| MixUp | 73.67 | 71.84 | 70.80 | 69.07 | 66.75 | 63.52 | 58.21 | 48.68 | 28.72 | 8.01 |
| CutMix | 76.13 | 73.77 | 72.93 | 71.92 | 70.32 | 68.32 | 65.48 | 59.30 | 43.11 | 17.04 |
| FMix | 71.27 | 68.52 | 66.49 | 65.78 | 65.00 | 62.31 | 57.80 | 48.98 | 30.09 | 6.19 |
| GridMix | 67.99 | 66.48 | 65.60 | 64.21 | 62.54 | 60.40 | 56.98 | 50.02 | 34.62 | 9.74 |
| ResizeMix | 66.69 | 66.90 | 62.80 | 58.15 | 47.00 | 36.15 | 36.49 | 33.07 | 10.97 | 2.06 |
| SaliencyMix | 73.50 | 72.12 | 71.88 | 71.21 | 69.97 | 67.62 | 63.03 | 53.12 | 33.27 | 11.06 |
| Attentive-CutMix | 78.26 | 72.49 | 67.52 | 63.83 | 60.76 | 54.93 | 38.05 | 27.75 | 32.20 | 31.66 |
| PuzzleMix | 78.01 | 76.42 | 75.54 | 74.61 | 73.99 | 71.46 | 68.23 | 63.24 | 51.51 | 26.25 |
| AutoMix | 77.52 | 76.74 | 75.14 | 72.95 | 69.71 | 66.76 | 62.59 | 59.00 | 49.01 | 27.45 |
| TransMix | 78.37 | 76.29 | 75.86 | 75.58 | **74.79** | **73.19** | **70.81** | **68.18** | 62.29 | 51.09 |
| DeiT | 77.50 | 75.99 | 75.80 | 74.57 | 73.80 | 71.73 | 67.23 | 58.40 | 43.49 | 19.94 |
| SMMix | 79.32 | **77.84** | **76.60** | 75.41 | 73.00 | 68.10 | 58.89 | 45.07 | 23.25 | 4.59 |
| MixPro | 79.91 | **78.37** | 75.36 | 72.08 | 68.37 | 57.48 | 37.08 | 21.62 | 11.87 | 3.69 |
| MergeMix | **81.29** | 75.38 | 73.41 | 69.36 | 61.02 | 51.73 | 52.52 | 62.14 | **64.83** | **57.21** |
| MergeMix[tome] | **81.66** | 77.45 | **78.25** | **78.26** | 74.85 | **72.02** | **69.70** | **67.81** | 67.80 | 60.56 |

## C.2 ROBUSTNESS EXPERIMENTS OF MIXUP AUGMENTATIONS

**Calibration of mixup augmentations** Table A3 shows the results of calibration on seven mixup augmentations. We evaluated with two public datasets by DeiT-Tiny, ViT-Small, and ViT-Large on the CIFAR100 dataset, by models trained with 200 epochs, DeiT-Tiny and ViT-Small trained with 600 epochs, and DeiT-Small on the ImageNet-1K dataset, trained with 300 epochs.

Table A6: The Top-1 accuracy of DeiT-Small trained by various Mixup approaches on the ImageNet-1K dataset with different occlusion ratios. tome denotes inference with token merging.

| Method | DeiT-Samll Trained 300 epochs | | | | | | | | | |
|---|---|---|---|---|---|---|---|---|---|---|
| | 0% | 10% | 20% | 30% | 40% | 50% | 60% | 70% | 80% | 90% |
| Vanilla | 75.46 | 74.03 | 72.85 | 71.36 | 69.91 | 67.94 | 64.71 | 60.42 | 51.65 | 34.08 |
| MixUp | 78.74 | 78.17 | 77.11 | 76.21 | 74.31 | 71.85 | 68.13 | 62.40 | 51.45 | 33.08 |
| CutMix | 80.16 | 79.23 | 78.20 | 76.87 | 75.21 | 72.92 | 69.24 | 64.63 | 55.95 | 39.69 |
| TransMix | 80.36 | 79.47 | 78.24 | 77.00 | 75.40 | 73.31 | 70.17 | 65.78 | 57.79 | 42.44 |
| DeiT | 80.27 | 79.03 | 77.92 | 76.39 | 74.65 | 72.25 | 68.31 | 63.53 | 54.57 | 38.81 |
| SMMix | 79.32 | 78.56 | 77.70 | 76.17 | 73.76 | 70.25 | 64.95 | 58.94 | 48.69 | 34.37 |
| MixPro | 79.25 | 78.83 | 78.01 | 77.24 | 76.02 | 74.26 | 71.48 | 67.44 | 58.49 | 39.47 |
| MergeMix | **80.70** | **80.38** | **79.95** | **78.97** | **78.08** | **76.21** | **74.08** | **70.71** | **65.57** | **51.98** |
| MergeMix$^{tome}$ | 79.67 | 79.57 | 79.01 | 78.60 | 77.68 | 76.11 | 73.99 | 70.56 | 64.84 | 51.80 |

Table A7: Classification results of CAFormer small (CAFormer-S12) with different mxiup augmentations, training 200 epochs with 100 batch size on CIFAR100 dataset.

| Model | Dataset | Epochs | Vanila | MixUp | CutMix | DeiT | TransMix | MergeMix |
|---|---|---|---|---|---|---|---|---|
| CAFormer-S12 | CIFAR100 | 200 | 74.95 | 81.64 | **84.69** | 83.60 | 83.70 | 84.30 |

**Results of occlusion robustness** The full results of occlusion robustness classification on MergeMix and other mixup methods. Figure A1 shows the curve of MergeMix and other mixup methods on the CIFAR100 dataset and ImageNet-1K dataset. Table A4 and Table A5 show the accuracy results on the CIFAR100 dataset by vanilla and 14 different mixup approaches. Table A6 shows the results of 8 different methods on the ImageNet-1K dataset.

## C.3  FURTHER RESULTS ON DIFFERENT BACKBONE AND DATA MODALITY.

For further exploring the effectiveness of MergeMix, we applied our approach on a hyper-model MetaFormer (CAFormer) (Yu et al., 2022) and two audio datasets for classification by HuBERT-Base (Hsu et al., 2021). ESC-50 (Piczak, 2015) consisted of 50 classes, with 1,200 training samples and 400 validation samples, and a maximum duration of 3 seconds. UrbanSound8k (Salamon et al., 2014) is a classification dataset consisting of 10 classes, with

Table A8: Supervised fine-tuning on HuBERT-Base model on ESC-50 and UbranSound8K datasets.

| HuBERT-Base | ESC-50 | UbranSound8K |
|---|---|---|
| Vanilla | 75.12±1.07 | 84.14±0.45 |
| MixUp | 75.86±0.83 | 85.02±0.26 |
| TransMix | 76.27±1.14 | 85.33±0.57 |
| MergeMix | 76.51±0.95 | 85.69±0.42 |

a maximum duration of 4 seconds, containing 7,079 training samples and 816 validation samples. Following the USB experimental setup, we fine-tuned the model for 100 epochs using the AdamW optimizer. The base learning rate was set to `1e-4`, texttt5e-4, with a batch size of 32 and a weight decay coefficient of 5e-4. Both the compared shuffling methods and our proposed MergeMix can be directly transferred to audio data (treated as one-dimensional sequences). Table A8 below shows our reproduced comparison results on two audio datasets. Compared to the MixUp and TransMix baselines, MergeMix achieves significant performance improvements over multiple shuffling baseline models. For CAFormer-S12, we train for 200 epochs on the CIFAR100 dataset with the same settings as DeiT-Tiny. Table A7 shows that MergeMix obtained a second-best result, since CAFormer only has with Attention module of 2 stages (8 layers).

# D  VISUAL UNDERSTANDING

## D.1  CALIBRATION RESULTS OF MLLM

To explore MLLM calibration, we selected 4 tasks with short responses across different token-reduction settings. We evaluate three scenarios: using full vision tokens, 50% tokens, and 25% tokens, and compare various data augmentation strategies (MixUp, CutMix, ResizeMix, MergeMix) trained with ranking loss (denoted as **rl**). Table A9 shows that with unfreezing the vision encoder, the ECE can be better than freezing. With the vision tokens reduced in training. Overall, token reduction leads to a moderate drop in calibration accuracy, but effective augmentation strategies significantly mitigate this degradation. In the full-token setting, CutMix$^{rl}$ achieves the lowest GQA calibration error (6.09), while ResizeMix$^{rl}$ shows the best SEED result (36.08). When reducing tokens to 50%, CutMix$^{rl}$ and MergeMix$^{rl}$ remain competitive, maintaining strong calibration across tasks despite reduced visual information. Even with 25% tokens, CutMix$^{rl}$ continues to yield relatively balanced performance, indicating that appropriate augmentation enhances robustness under token compression. These results suggest that MLLM calibration remains an open question, especially in environments that require a reliable answer.

Table A9: The calibration results of LLaVA on POPE, ScienceVQA$^I$, GQA & SEED$^I$.

| Method | GQA | POPE | SEED$^I$ | ScienceVQA$^I$ |
|---|---|---|---|---|
| Baseline | 14.57 | 13.16 | 33.79 | 28.09 |
| **Training with Full Vision Tokens** | | | | |
| SFT Vision | 8.52 | 12.82 | 32.67 | 21.66 |
| +MixUp$^{rl}$ | 6.09 | 12.72 | 33.26 | 21.51 |
| +CutMix$^{rl}$ | 6.74 | 12.62 | 32.77 | 24.71 |
| +ResizeMix$^{rl}$ | 12.53 | 13.17 | 36.08 | 24.58 |
| +MergeMix$^{rl}$ | 6.50 | 12.91 | 32.52 | 23.66 |
| **Training with 50% Vision Tokens** | | | | |
| SFT Vision | 18.13 | 12.67 | 34.41 | 24.28 |
| +MixUp$^{rl}$ | 13.40 | 12.74 | 33.60 | 22.61 |
| +CutMix$^{rl}$ | 10.48 | 12.67 | 33.83 | 20.63 |
| +ResizeMix$^{rl}$ | 12.60 | 12.97 | 37.41 | 23.74 |
| +MergeMix$^{rl}$ | 10.34 | 12.76 | 33.37 | 25.22 |
| **Training with 25% Vision Tokens** | | | | |
| SFT Vision | 13.32 | 12.51 | 34.97 | 18.86 |
| +MixUp$^{rl}$ | 12.97 | 12.66 | 34.89 | 19.33 |
| +CutMix$^{rl}$ | 12.23 | 13.10 | 34.85 | 20.70 |
| +ResizeMix$^{rl}$ | 10.66 | 14.17 | 38.77 | 17.27 |
| +MergeMix$^{rl}$ | 12.55 | 12.17 | 34.87 | 17.70 |

## D.2  RELATIONSHIP BETWEEN MIXING RATIOS AND REWARDS

To understand how the proposed ranking loss on synthetic mixed pairs approximates human preference learning, we conduct a `VLM-as-the-judge-eval` evaluation in which strong MLLMs score the mixed pairs generated with different mixing ratios $\lambda$. Specifically, we query a diverse set of frontier MLLMs, including Grok-3, Doubao-Seed-1.6 (Guo et al., 2025), Doubao-Seed-1.6-CoT, Qwen3-VL-Plus-32B (Yang et al., 2025a), Qwen2.5-VL-72B (Bai et al., 2025), Gemini-2.5 (Team et al., 2023), and a human-expert baseline, to assign reward scores to each mixed pair. As shown in Figure A2, we found that the reward consistently increases with larger mixing ratios, and this trend holds across nearly all evaluators. The strong monotonic correlation suggests that $\lambda$ provides a reliable and well-behaved control signal for modeling preference strength. This observation validates our use of $\lambda$ as an interpretable proxy for preference supervision and highlights its potential as a lightweight alternative to explicit human-annotated reward signals.

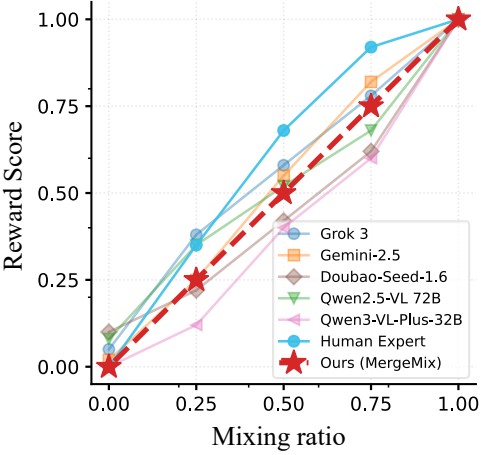

Figure A2: **VLM-as-the-Judge results for different mixing ratios.** Higher mixing ratios consistently yield higher reward scores across strong MLLMs. MergeMix aligns closely with these models, showing that the mixing ratio provides a meaningful preference signal.

Table A10: **Different token merge ratios of inference comparison results with augmentations**. **AVG**: The average of the nine benchmarks for comprehensive comparison, except for MME.

| LLaVA-7b v1.5 | Train Ratio | Image Question Answering | | | | | Benchmarks | | | | | AVG | Gain |
|---|---|---|---|---|---|---|---|---|---|---|---|---|---|
| | | VQAv2 | GQA | VizWiz | SciVQA$^I$ | TextVQA | MME | MMBench | MMBench$^{CN}$ | POPE | SEED$^I$ | | |
| **Inference Full Vision Token** | | | | | | | | | | | | | |
| Vanilla | – | 78.5 | 62.0 | 50.0 | 66.8 | 58.2 | 1510.7 | 64.3 | 58.3 | 85.87 | 66.19 | 65.57 | – |
| SFT Vision | 100% | 79.32 | 62.98 | 47.45 | 70.05 | 57.17 | 1490.88 | 66.26 | 60.05 | 86.18 | 67.32 | 66.31 | **+0.74** |
| + MixUp | 100% | 79.27 | 62.58 | 44.95 | 69.41 | 57.39 | 1483.20 | 65.72 | 58.24 | 86.27 | 66.73 | 65.62 | **+0.05** |
| + CutMix | 100% | 79.18 | 62.40 | 45.04 | 70.60 | 57.06 | 1452.31 | 66.32 | 58.24 | 86.47 | 67.22 | 65.84 | **+0.27** |
| + ResizeMix | 100% | 77.78 | 61.66 | 44.43 | 68.91 | 55.11 | 1436.09 | 63.91 | 55.41 | 86.01 | 63.91 | 64.13 | **-1.44** |
| + MergeMix | 100% | 79.24 | 62.44 | 47.69 | 69.86 | 57.56 | 1479.97 | 66.58 | 60.65 | 86.10 | 67.47 | 66.40 | **+0.83** |
| **Inference 75% Vision Token** | | | | | | | | | | | | | |
| Vanilla | – | 77.24 | 59.65 | 50.86 | 68.42 | 55.66 | 1460.88 | 63.05 | 57.9 | 85.60 | 65.21 | 64.84 | – |
| SFT Vision | 100% | 77.62 | 59.73 | 46.57 | 70.15 | 54.83 | 1454.99 | 65.03 | 59.36 | 85.60 | 65.90 | 64.98 | **+0.14** |
| + MixUp | 100% | 77.66 | 59.37 | 44.15 | 69.91 | 56.18 | 1457.98 | 64.77 | 57.73 | 85.15 | 65.28 | 64.47 | **-0.37** |
| + CutMix | 100% | 77.67 | 59.21 | 44.25 | 69.66 | 54.84 | 1400.49 | 65.03 | 57.98 | 85.91 | 65.66 | 64.47 | **-1.09** |
| + ResizeMix | 100% | 76.45 | 59.65 | 43.31 | 68.82 | 53.14 | 1426.75 | 63.91 | 55.15 | 85.29 | 63.04 | 63.20 | **-1.64** |
| + MergeMix | 100% | 77.71 | 59.32 | 47.46 | 70.70 | 54.85 | 1440.50 | 65.37 | 58.93 | 85.04 | 65.98 | 65.04 | **+0.20** |
| **Inference 50% Vision Token** | | | | | | | | | | | | | |
| Vanilla | – | 76.65 | 59.33 | 50.45 | 68.86 | 55.33 | 1452.18 | 62.8 | 56.87 | 86.53 | 64.09 | 64.55 | – |
| SFT Vision | 100% | 77.07 | 59.05 | 46.19 | 70.35 | 54.40 | 1436.79 | 64.60 | 58.93 | 85.62 | 65.12 | 64.59 | **+0.04** |
| + MixUp | 100% | 77.11 | 58.96 | 44.39 | 69.36 | 54.41 | 1422.28 | 64.34 | 57.90 | 86.16 | 64.29 | 64.10 | **-0.45** |
| + CutMix | 100% | 77.15 | 58.66 | 44.00 | 69.31 | 54.78 | 1428.21 | 64.94 | 59.02 | 86.42 | 64.80 | 64.34 | **-0.21** |
| + ResizeMix | 100% | 75.90 | 59.46 | 42.99 | 69.31 | 52.83 | 1444.13 | 63.23 | 54.20 | 85.77 | 62.07 | 62.86 | **-1.69** |
| + MergeMix | 100% | 77.13 | 59.02 | 47.46 | 70.55 | 54.54 | 1461.63 | 65.03 | 58.84 | 85.70 | 65.25 | 64.84 | **+0.29** |
| **Inference 25% Vision Token** | | | | | | | | | | | | | |
| Vanilla | – | 74.63 | 58.76 | 52.71 | 68.67 | 55.32 | 1398.24 | 60.65 | 54.03 | 86.54 | 62.05 | 63.71 | – |
| SFT Vision | 100% | 75.02 | 58.43 | 46.36 | 69.06 | 52.32 | 1376.88 | 62.37 | 55.84 | 85.64 | 63.63 | 63.19 | **-0.52** |
| + MixUp | 100% | 75.45 | 58.63 | 44.82 | 68.02 | 52.13 | 1384.31 | 62.28 | 60.65 | 85.54 | 62.97 | 63.39 | **-0.32** |
| + CutMix | 100% | 75.39 | 58.61 | 43.85 | 68.67 | 52.69 | 1330.99 | 62.71 | 56.01 | 86.30 | 63.02 | 63.03 | **-0.68** |
| + ResizeMix | 100% | 74.08 | 59.02 | 43.40 | 68.96 | 51.40 | 1377.96 | 61.08 | 53.09 | 86.53 | 59.85 | 61.93 | **-1.78** |
| + MergeMix | 100% | 75.50 | 58.60 | 48.14 | 69.86 | 52.01 | 1439.44 | 63.05 | 57.56 | 85.53 | 63.72 | 63.77 | **+0.06** |

## D.3 RESULTS OF DIFFERENT VISION TOKEN RATIOS ON INFERENCE

In this subsection, we validate the different ratios of vision tokens on the LLaVA benchmark. Table A10, Table A11 and Table A12 show the fully results with full, 75%, 50% and 25% ratios on inference, respectively. Those results give a full comparison of the influence on the vision tokens. Significantly shown in Table A10, MergMix always brings gains in different merge ratios, from **+0.83** to **+0.06**. Other methods, since they are highly random, cause performance instability.

From the results shown in Table A11, when the training stage uses the token merge, it can achieve an average gain of **66.84%**, which improves **0.43%** over training and inference without token merge training and inference, with an improvement of **+1.27%** than the average performance of the original LLaVA model. Figure A3 shows the average accuracy on the LLaVA benchmark for the LLaVA-v1.5-7B model trained and evaluated under different vision token-merging ratios. The results demonstrate that MergeMix maintains strong performance across a wide range of settings, outperforming or matching other baselines.

## D.4 RESULTS OF DIFFERENT RANKING LOSS

To understand the effectiveness of ranking loss on preference tuning, we conducted an ablation study for $\mathcal{L}_{\text{SimPo}}^{\text{Mix}}$. Table A13 shows that compared with vanilla SimPO, our approach can bring more improvement.

Table A11: **Different token merge ratios of inference comparison results with augmentations**. **AVG**: The average of the nine benchmarks for comprehensive comparison, except for MME.

| LLaVA-7b v1.5 | Train Ratio | Image Question Answering | | | | | Benchmarks | | | | | AVG | Gain |
|---|---|---|---|---|---|---|---|---|---|---|---|---|---|
| | | VQAv2 | GQA | VizWiz | SciVQA$^T$ | TextVQA | MME | MMBench | MMBench$^{CN}$ | POPE | SEED$^I$ | | |
| **Inference Full Vision Token** | | | | | | | | | | | | | |
| Vanilla | − | 78.5 | 62.0 | 50.0 | 66.8 | 58.2 | 1510.7 | 64.3 | 58.3 | 85.87 | 66.19 | 65.57 | − |
| SFT Vision | 50% | 78.6 | 62.47 | 48.15 | 69.51 | 56.41 | 1486.24 | 66.32 | 57.98 | 87.37 | 66.75 | 65.95 | **+0.38** |
| + MixUp | 50% | 78.51 | 62.07 | 51.1 | 68.47 | 56.54 | 1459.06 | 65.63 | 59.53 | 86.86 | 66.06 | 66.08 | **+0.51** |
| + CutMix | 50% | 78.58 | 62.39 | 50.53 | 70.2 | 55.95 | 1414.72 | 66.92 | 59.53 | 86.56 | 66.2 | 66.31 | **+0.74** |
| + ResizeMix | 50% | 76.39 | 61.05 | 45.48 | 68.07 | 54.60 | 1447.35 | 63.31 | 51.97 | 86.57 | 62.54 | 63.33 | **-2.24** |
| + MergeMix | 50% | 78.61 | 62.18 | 52.14 | 69.61 | 56.85 | 1453.97 | 66.58 | 59.02 | 86.47 | 66.63 | 66.45 | **+0.88** |
| **Inference 75% Vision Token** | | | | | | | | | | | | | |
| Vanilla | − | 77.24 | 59.65 | 50.86 | 68.42 | 55.66 | 1460.88 | 63.05 | 57.90 | 85.60 | 65.21 | 64.84 | − |
| SFT Vision | 50% | 78.75 | 62.82 | 48.02 | 70.65 | 56.33 | 1486.24 | 66.40 | 59.02 | 86.93 | 67.17 | 66.23 | **+1.39** |
| + MixUp | 50% | 78.87 | 62.32 | 51.01 | 69.11 | 56.62 | 1480.04 | 65.63 | 59.53 | 86.86 | 66.06 | 66.22 | **+1.38** |
| + CutMix | 50% | 78.73 | 62.42 | 49.85 | 70.50 | 56.12 | 1418.07 | 67.61 | 59.87 | 85.96 | 66.44 | 66.39 | **+1.55** |
| + ResizeMix | 50% | 76.79 | 61.12 | 44.85 | 68.37 | 54.24 | 1475.27 | 64.26 | 52.66 | 85.67 | 63.30 | 63.47 | **-1.37** |
| + MergeMix | 50% | 78.81 | 62.50 | 52.31 | 69.56 | 56.51 | 1455.81 | 66.66 | 59.10 | 85.76 | 67.12 | 66.48 | **+1.64** |
| **Inference 50% Vision Token** | | | | | | | | | | | | | |
| Vanilla | − | 76.65 | 59.33 | 50.45 | 68.86 | 55.33 | 1452.18 | 62.80 | 56.87 | 86.53 | 64.09 | 64.55 | − |
| SFT Vision | 50% | 78.49 | 63.39 | 46.69 | 70.25 | 55.68 | 1468.38 | 66.83 | 57.76 | 86.47 | 66.48 | 65.78 | **+1.23** |
| + MixUp | 50% | 78.54 | 61.91 | 51.01 | 69.61 | 55.76 | 1468.14 | 65.63 | 59.87 | 86.51 | 66.39 | 66.14 | **+1.59** |
| + CutMix | 50% | 78.50 | 62.18 | 48.79 | 70.50 | 55.83 | 1431.32 | 67.18 | 59.02 | 86.00 | 66.27 | 66.03 | **+1.48** |
| + ResizeMix | 50% | 76.55 | 60.79 | 44.20 | 68.32 | 54.21 | 1470.22 | 63.31 | 52.66 | 86.23 | 62.73 | 63.22 | **-1.33** |
| + MergeMix | 50% | 78.51 | 62.09 | 51.01 | 70.10 | 56.03 | 1464.00 | 66.75 | 59.45 | 86.05 | 66.39 | 66.26 | **+1.71** |
| **Inference 25% Vision Token** | | | | | | | | | | | | | |
| Vanilla | − | 74.63 | 58.76 | 52.71 | 68.67 | 55.32 | 1398.24 | 60.65 | 54.03 | 86.54 | 62.05 | 63.71 | − |
| SFT Vision | 50% | 77.30 | 61.77 | 46.33 | 70.55 | 54.19 | 1411.01 | 65.72 | 58.07 | 86.34 | 65.46 | 65.08 | **+1.37** |
| + MixUp | 50% | 77.28 | 61.56 | 48.77 | 69.56 | 54.10 | 1419.71 | 66.32 | 56.27 | 86.57 | 65.32 | 65.08 | **+1.37** |
| + CutMix | 50% | 77.20 | 61.52 | 49.00 | 71.24 | 53.96 | 1372.66 | 66.49 | 58.76 | 86.38 | 65.24 | 65.53 | **+1.82** |
| + ResizeMix | 50% | 75.06 | 59.88 | 43.12 | 67.13 | 52.34 | 1445.26 | 61.51 | 51.63 | 86.03 | 61.59 | 62.03 | **-1.68** |
| + MergeMix | 50% | 77.20 | 61.81 | 51.66 | 70.35 | 54.47 | 1401.58 | 66.83 | 59.10 | 85.92 | 65.44 | 66.84 | **+3.13** |

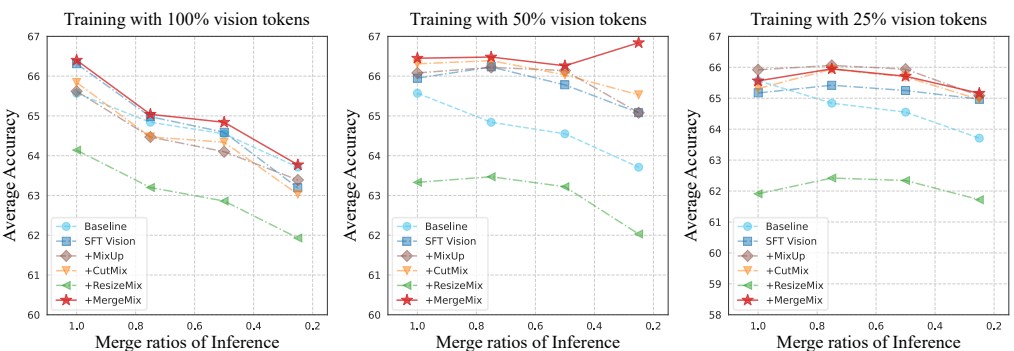

Figure A3: The plots of the LLaVA-v1.5-7B model under different inference time merge ratios for various methods (Baseline, SFT, MixUp, CutMix, ResizeMix, and MergeMix) demonstrate that MergeMix maintains robust performance across a wide range of configurations.

# E  EFFICIENCY

## E.1  RESULTS OF DIFFERENT VISION TOKEN RATIOS ON INFERENCE

To further validate the inference efficiency gains achieved by Token Merge in MergeMix, we conducted experiments on both image classification models and multi-modal large models. As shown in Table A14, increasing the merging ratio $r$ of the ViT-L model significantly reduces FLOPs (from 59.57G to **34.93**G) while throughput improves from 122.83 to **201.07** (**+63.7%**). Moreover, the

Table A12: Different token merge ratios of inference comparison results with augmentations. **AVG**: The average of the nine benchmarks for comprehensive comparison, except for MME.

| LLaVA-7b v1.5 | Train Ratio | Image Question Answering | | | | | Benchmarks | | | | | AVG | Gain |
|---|---|---|---|---|---|---|---|---|---|---|---|---|---|
| | | VQAv2 | GQA | VizWiz | SciVQA$^I$ | TextVQA | MME | MMBench | MMBench$^{CN}$ | POPE | SEED$^I$ | | |
| **Inference Full Vision Token** | | | | | | | | | | | | | |
| Vanilla | − | 78.5 | 62.0 | 50.0 | 66.8 | 58.2 | 1510.7 | 64.3 | 58.3 | 85.87 | 66.19 | 65.57 | − |
| SFT Vision | 25% | 77.92 | 62.01 | 50.30 | 69.11 | 55.03 | 1420.69 | 64.17 | 56.09 | 86.82 | 65.04 | 65.17 | **-0.40** |
| + MixUp | 25% | 77.89 | 62.01 | 52.53 | 70.20 | 55.90 | 1444.45 | 64.29 | 57.73 | 86.94 | 65.77 | 65.92 | **+0.35** |
| + CutMix | 25% | 77.92 | 61.57 | 51.02 | 69.21 | 55.43 | 1408.18 | 64.23 | 57.13 | 86.09 | 65.19 | 65.31 | **-0.26** |
| + ResizeMix | 25% | 75.38 | 59.77 | 41.38 | 66.78 | 53.45 | 1430.64 | 62.37 | 51.54 | 85.26 | 61.26 | 61.91 | **-3.66** |
| + MergeMix | 25% | 77.86 | 61.54 | 50.50 | 69.56 | 55.40 | 1458.49 | 64.86 | 57.98 | 87.22 | 65.10 | 65.56 | **-0.01** |
| **Inference 75% Vision Token** | | | | | | | | | | | | | |
| Vanilla | − | 77.24 | 59.65 | 50.86 | 68.42 | 55.66 | 1460.88 | 63.05 | 57.9 | 85.60 | 65.21 | 64.84 | − |
| SFT Vision | 25% | 78.09 | 62.11 | 49.67 | 69.86 | 55.03 | 1423.27 | 64.94 | 57.47 | 85.86 | 65.77 | 65.42 | **+0.58** |
| + MixUp | 25% | 77.97 | 61.23 | 53.13 | 69.46 | 56.17 | 1466.33 | 66.06 | 58.67 | 86.14 | 65.67 | 66.06 | **+1.22** |
| + CutMix | 25% | 78.11 | 61.85 | 50.46 | 69.71 | 56.08 | 1420.67 | 67.52 | 59.19 | 85.14 | 65.42 | 65.94 | **+1.1** |
| + ResizeMix | 25% | 76.01 | 59.95 | 41.94 | 66.63 | 53.75 | 1459.84 | 63.48 | 52.92 | 84.99 | 62.08 | 62.42 | **-2.42** |
| + MergeMix | 25% | 78.07 | 61.42 | 50.17 | 70.2 | 55.96 | 1483.82 | 66.58 | 59.45 | 86.18 | 65.49 | 65.95 | **+1.11** |
| **Inference 50% Vision Token** | | | | | | | | | | | | | |
| Vanilla | − | 76.65 | 59.33 | 50.45 | 68.86 | 55.33 | 1452.18 | 62.8 | 56.87 | 86.53 | 64.09 | 64.55 | − |
| SFT Vision | 25% | 77.99 | 61.77 | 48.79 | 70.10 | 54.91 | 1443.02 | 64.69 | 57.38 | 86.21 | 65.41 | 65.25 | **+0.7** |
| + MixUp | 25% | 77.89 | 61.68 | 50.85 | 70.05 | 56.0 | 1448.62 | 66.92 | 58.07 | 86.44 | 65.54 | 65.94 | **+1.39** |
| + CutMix | 25% | 77.32 | 61.62 | 48.86 | 69.96 | 55.97 | 1428.02 | 67.01 | 59.02 | 85.89 | 65.57 | 65.69 | **+1.14** |
| + ResizeMix | 25% | 75.78 | 59.95 | 40.82 | 67.33 | 53.44 | 1456.66 | 63.4 | 53.09 | 85.57 | 61.65 | 62.34 | **-2.21** |
| + MergeMix | 25% | 77.91 | 61.56 | 48.86 | 70.5 | 56.0 | 1477.47 | 66.15 | 58.76 | 86.41 | 65.19 | 65.71 | **+1.16** |
| **Inference 25% Vision Token** | | | | | | | | | | | | | |
| Vanilla | − | 74.63 | 58.76 | 52.71 | 68.67 | 55.32 | 1398.24 | 60.65 | 54.03 | 86.54 | 62.05 | 63.71 | − |
| SFT Vision | 25% | 76.97 | 61.43 | 49.79 | 70.0 | 53.56 | 1405.86 | 64.94 | 56.7 | 86.81 | 64.51 | 64.97 | **+1.26** |
| + MixUp | 25% | 76.84 | 61.15 | 49.31 | 69.66 | 53.98 | 1409.16 | 65.8 | 57.81 | 86.72 | 64.49 | 65.08 | **+1.37** |
| + CutMix | 25% | 76.89 | 61.3 | 48.21 | 69.32 | 53.99 | 1370.27 | 66.32 | 57.98 | 86.42 | 64.21 | 64.96 | **+1.25** |
| + ResizeMix | 25% | 75.01 | 59.72 | 40.82 | 66.78 | 51.66 | 1418.27 | 62.45 | 52.66 | 85.87 | 60.54 | 61.72 | **-1.99** |
| + MergeMix | 25% | 76.91 | 61.0 | 48.79 | 70.2 | 54.51 | 1441.07 | 66.06 | 58.24 | 86.57 | 64.2 | 65.16 | **+1.45** |

Table A13: Ablation study of different ranking loss on LLaVA-v1.5-7B.

| Models | SciQA$^I$ | TextVQA | VizWiz | MMBench | Avg. | Gains |
|---|---|---|---|---|---|---|
| LLaVA-v1.5-7B | 66.8 | 58.2 | 50.0 | 64.3 | 59.82 | − |
| mDOP (Wang et al., 2024a) | 67.53 | 57.90 | 50.04 | 64.60 | 60.02 | **+0.20** |
| Re-Align (Xing et al., 2025) | 68.10 | **58.55** | **50.06** | 64.69 | 60.35 | **+0.53** |
| vanilla SimPO | 69.86 | 56.62 | 49.26 | 66.24 | 60.49 | **+0.67** |
| + MergeMix | **69.86** | 57.56 | 47.69 | **66.58** | 61.29 | **+1.47** |

additional overhead of Token Merge itself is extremely low (only 0.97 ms at $r = 0.75$), far below the computational cost of pre-layering. This demonstrates that Token Merge can efficiently compress visual tokens while maintaining negligible overhead. For multi-modal inference, Table A15 shows the effectiveness of Token Merge on LLaVA-v1.5-7B. As $r$ increases, the model's overall throughput improves, while TTFT decreases significantly. The optimal trade-off is achieved at $r = 0.5$, boosting throughput from 45.96 to **47.28** and reducing TTFT from 86.44 ms to **66.74** ms. When $r = 0.75$, TTFT further decreases to 62.25 ms. This demonstrates that Token Merge effectively accelerates multi-modal inference without compromising image-text alignment quality. These results collectively show that Token Merge can efficiently reduce the number of visual tokens in MergeMix, thereby lowering FLOPs and inference latency in both visual models and MLLMs. This validates the necessity and advantages of introducing Token Merge into our methodology.

# F VISUALIZATION AND CASE STUDY

In this section, we provide a visualization of the case study of augmentation samples with corresponding performance reward scores and an extensive visualization of mixing samples and ToMe

Table A14: Results of throughput, FLOPs (G), overhead of pre-layer and ToMe by ViT-L model with different merged ratios, evaluated on an Nvidia A100 GPU.

| Ratios | Throughput ↑ | FLOPs (G) | Overhead Layer | Overhead ToMe |
|---|---|---|---|---|
| Baseline | 122.83 | 59.57 | 10.19 | — |
| $r = 0.1$ | 130.45 | 56.29 | 9.82 | 3.72 |
| $r = 0.25$ | 145.98 | 49.37 | 8.23 | 3.14 |
| $r = 0.5$ | 167.54 | 42.65 | 5.52 | 1.79 |
| $r = 0.75$ | **201.07** | **34.93** | **3.23** | **0.97** |

Table A15: Results of throughput and Time-To-First-Token (TTFT) in ms on LLaVA-v1.5-7B with different merged ratios $r$, which is evaluated on a Nvidia A100 GPU.

| Ratios | Throughput ↑ | TTFT (ms) |
|---|---|---|
| LLaVA-V1.5 | 45.96 | 86.44 |
| $r = 0.1$ | 43.94 | 89.32 |
| $r = 0.25$ | 46.99 | 83.65 |
| $r = 0.5$ | **47.28** | 66.74 |
| $r = 0.75$ | 46.92 | **62.25** |

source maps. Firstly, we provide a case of different degrees of augmentation in Figure A4. Then, we plot some visualizations of token merge with different merge ratios, mixed samples with different $\lambda$ in Figure A5 and Figure A6. For every three rows in Figure A5 and Figure A6, the first and second rows of source maps could directly capture the important regions of the raw images, where a large merge ratio enables better grouping of the similar regions. Based on the source maps, when the mixing ratio $\lambda$ grows from small to large, MergeMix could keep the most distinguishable tokens of the first image while gradually expanding more tokens from less important regions, which enables MergeMix to generate reliable mixing samples with no more computational costs. Moreover, we provide GradCAM (Selvaraju et al., 2019) visualization of the top-1 and top-2 classes with the mixed samples of MergeMix on ImageNet-1K, as shown in Figure A7 and Figure A8.

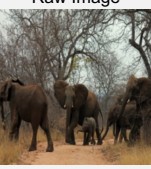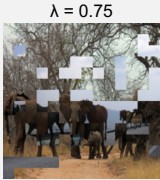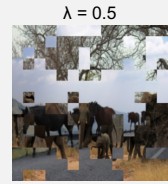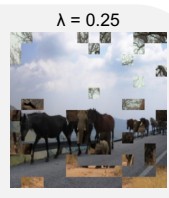
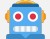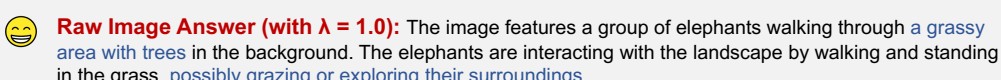

Figure A4: The visualization of the visual question answers with different mixing ratios by LLaVA-v1.5-7B model. Note that the blue texts denote the core question and the corresponding correct answers, while the green texts denote the wrong answer to the question. The raw image denotes without any augmentations, and other images denote with different mixing ratios $\lambda$. Ground-truth Answer denotes the raw labels for this case. With the mixing degree improving, the answer comes out more wrong or unrelated to the question, as shown in green color.

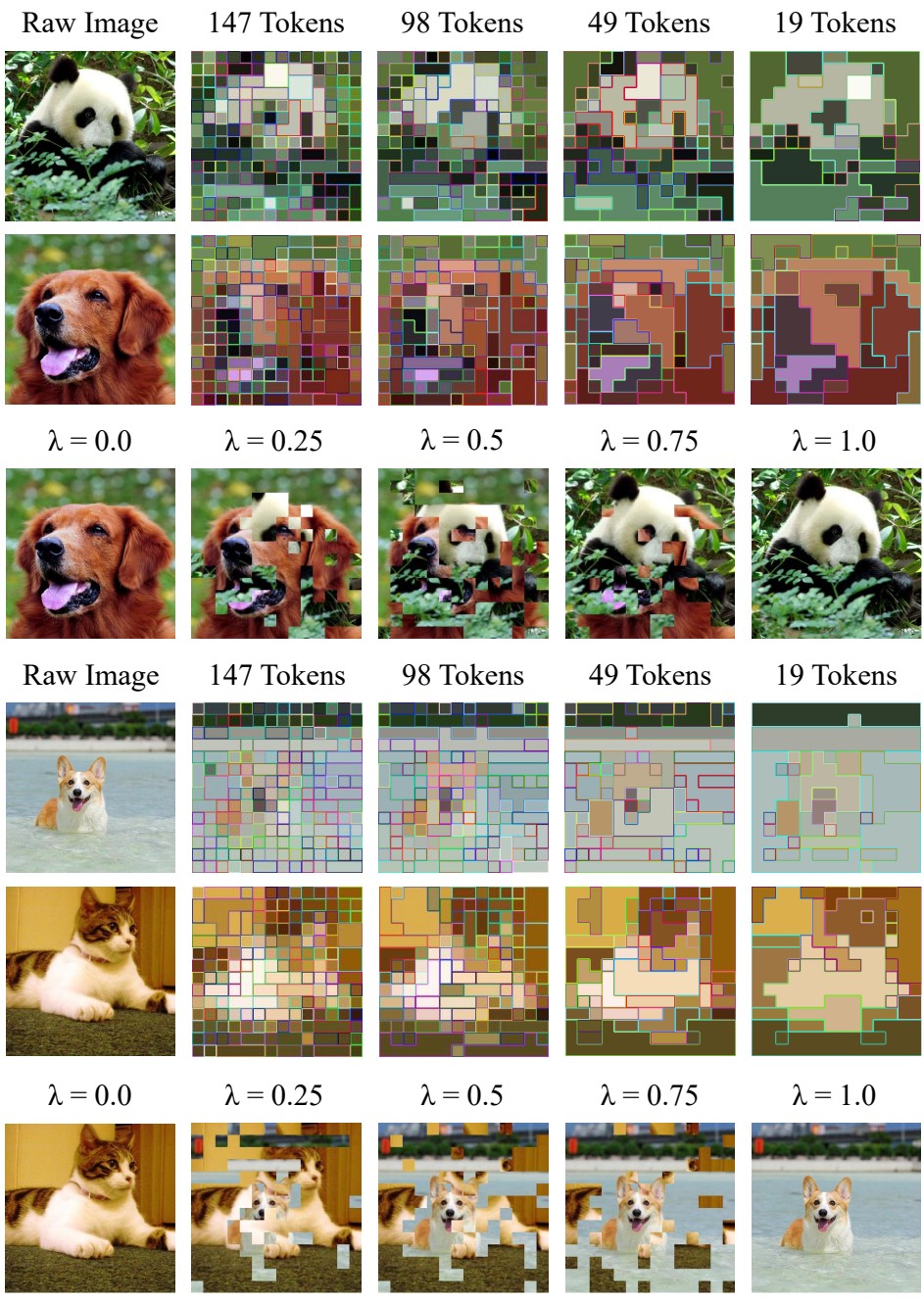

Figure A5: Visualization of mixed samples with source maps of ToMe with different mixing ratios $\lambda$ and various merge ratios on ImageNet-1K.

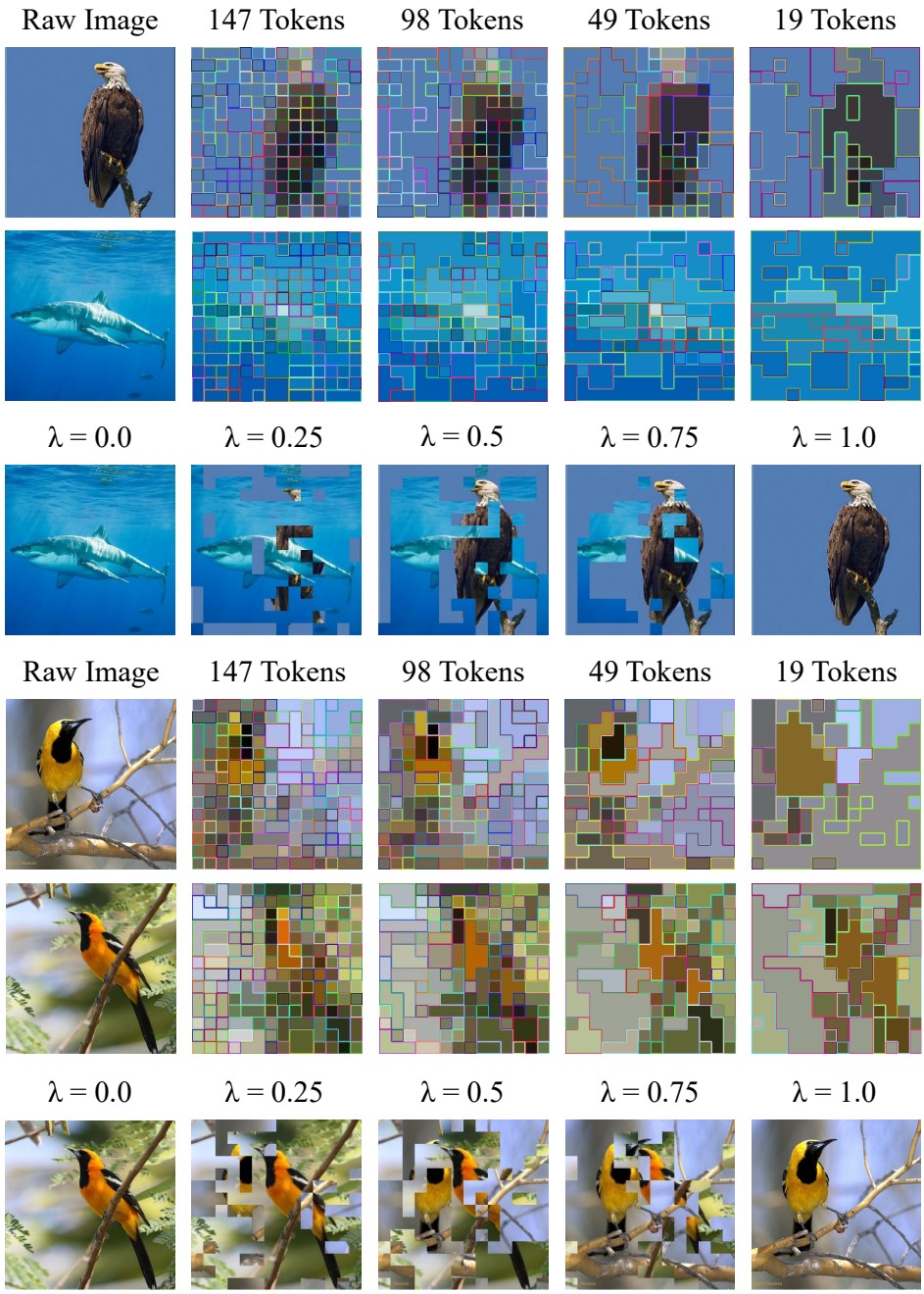

Figure A6: Visualization of mixed samples with source maps of ToMe with different mixing ratios $\lambda$ and various merge ratios on ImageNet-1K.

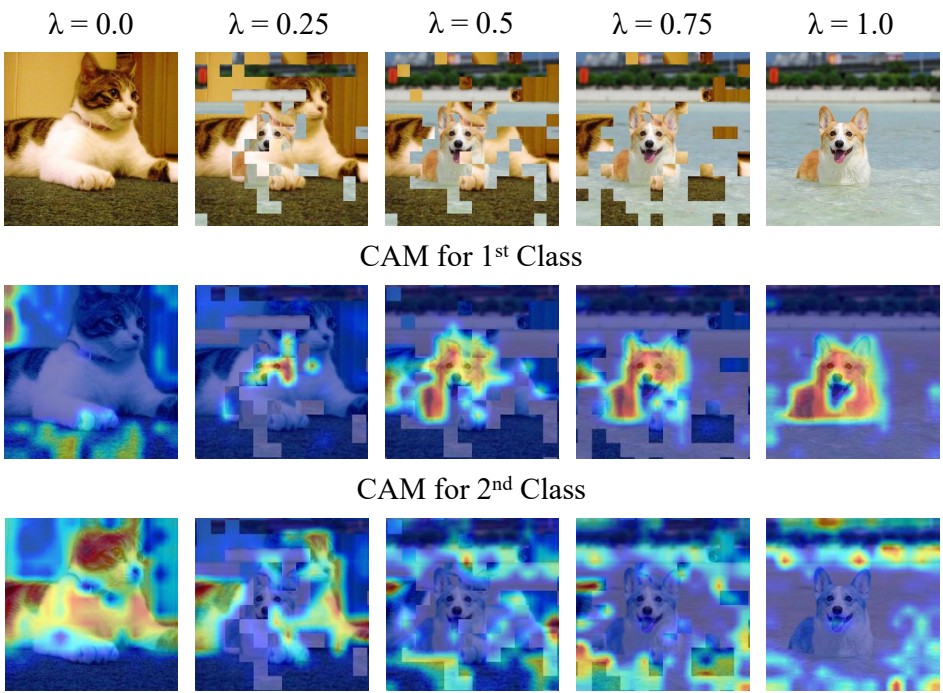

Figure A7: Visualization of mixed samples and corresponding GradCAM (Selvaraju et al., 2019) of the top-1/2 class with MergeMix on ImageNet-1K.

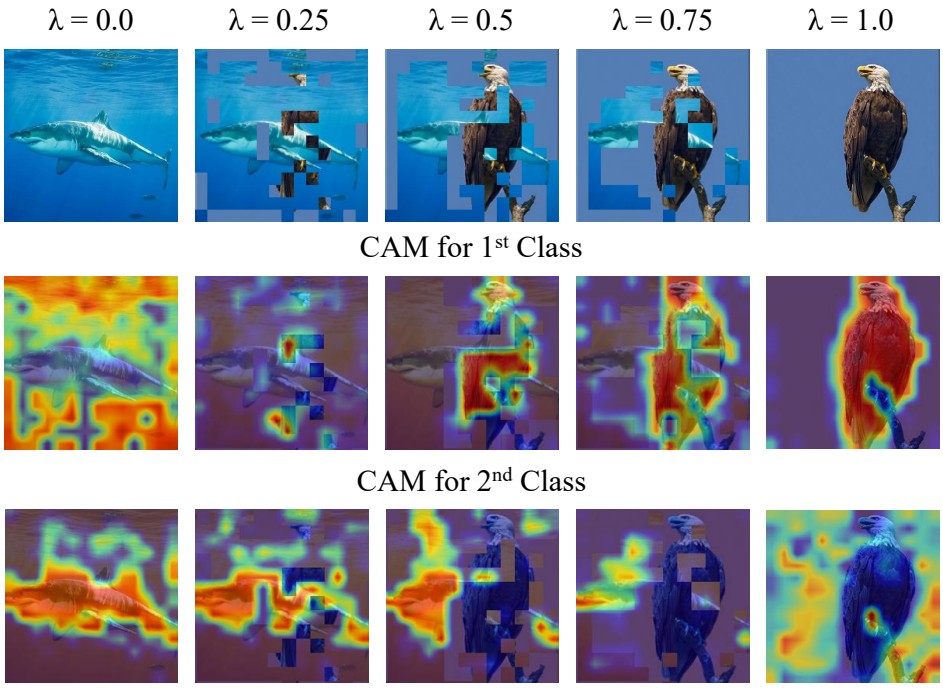

Figure A8: Visualization of mixed samples and corresponding GradCAM (Selvaraju et al., 2019) of the top-1/2 class with MergeMix on ImageNet-1K.

