# OpenReview forum: "MergeMix: A Unified Augmentation Paradigm for Visual and Multi-Modal Understanding"
_ICLR.cc/2026/Conference — ICLR 2026 Poster_

### Official Review · Reviewer_eiyX · 2025-10-29

**Soundness:** 3
**Presentation:** 3
**Contribution:** 3
**Rating:** 6
**Confidence:** 3

**Summary:**

This paper proposes MergeMix, a unified augmentation framework for visual and multimodal learning. It combines token merging and mixup to create attention-aware mixed images, which are then used to build preference pairs for training. The model treats clean samples as “winners” and mixed ones as “losers,” optimized with the SimPO loss to bridge supervised fine-tuning (SFT) and reinforcement learning (RL). Experiments on image classification (CIFAR100, ImageNet-1K, Stanford Cars) and MLLM benchmarks (LLaVA, Qwen2.5-VL) show that MergeMix improves accuracy, calibration, and training efficiency compared with existing mixup and alignment methods.

**Strengths:**

- The paper proposes a unified augmentation framework that connects supervised fine-tuning and preference optimization.

- The idea of using token merging to generate mixed samples is a reasonable extension of existing mixup methods and leverages attention information effectively.

- Experiments cover both image classification and multimodal benchmarks, showing consistent  improvements over baselines such as CutMix, MixPro, and SeVa.

- The method achieves competitive performance with lower computational cost and good training efficiency, suggesting potential for practical use.

**Weaknesses:**

- Incremental Overlap with Prior Work: While token merging and mixup are cleverly combined, the conceptual novelty may be perceived as moderate since both techniques are pre-existing; the innovation mainly lies in their integration.

- Marginal Multimodal Gains: Improvements on LLaVA benchmarks (0.8%) and Qwen2.5-VL (2.9%) are positive but not substantial, raising questions about statistical significance.

**Questions:**

- 1.Comparison with RL-based Preference Methods: Since MergeMix aims to bridge SFT and RL, it would be helpful to include direct comparisons with RL-based methods such as PPO or GRPO on MLLM benchmarks to quantify the alignment improvement.

- 2.Statistical Significance of Gains: Many reported improvements (e.g., +0.8% on LLaVA) are relatively small. Are these consistent across multiple random seeds? Please report standard deviations or significance tests to confirm reliability.

- 3.Generality Beyond ViT-based Models: MergeMix is mainly evaluated with ViT and DeiT architectures. Could the authors test it on convolutional backbones or hybrid models to verify broader applicability?

- 4.Computational Overhead of Token Merge: Although the paper claims better efficiency, the token merging and reconstruction steps may introduce overhead. Can the authors provide a detailed runtime breakdown to clarify the trade-offs?

- 5.Visualization and Qualitative Analysis: Including visual examples of merged attention maps or mixed images could help readers understand what semantic information MergeMix preserves compared to conventional mixup.

I will adjust the score based on the authors’ response.

---

> ### Author Response · Authors · 2025-11-22
> **Response to Reviewer eiyX [1/3]**
>
> ### **[W1] Incremental overlap with prior works.**
>
> **Response:**  We would like to clarify that MergeMix is *not* a simple juxtaposition of token merging and mixup. Prior works on token merging focus on efficiency, whereas mixup-based augmentations rely on metrics from total forward (e.g, TransMix, SMMix, and MixPro), instead of efficiency. None of them address how *data-dependent token saliency* should guide sample mixing in multimodal large language models (MLLMs), nor how mixing policies should adapt to cross-modal reasoning. As highlighted in the Introduction, our contribution lies in bridging this gap:
>
> 1. **We formulate a new mixing policy that uses attention-derived token saliency to determine $\hat{\lambda}$**, enabling an explicit trade-off between preservation of semantic anchors and synthetic diversity—something that neither classical mixup nor token merging provides.
>
> 2. **We extend saliency-aware augmentation to MLLMs**, where mixing must occur not in pixel space but in the latent token space, requiring alignment between visual tokens and language-conditioned reasoning.
>
> 3. **MergeMix unifies merging (efficiency) and mixing (performance) under the same pipeline**, allowing controllable generation of data-dependent synthetic samples for preference-based training.
>
> Thus, MergeMix goes beyond directly combining two existing techniques; it introduces a principled and MLLM-specific augmentation mechanism that cannot be achieved by prior components in isolation.
>
> ---
>
> ### **[W2] Marginal multimodal gains.**
>
> **Response:** We thank the reviewer for pointing this out. In contrast, the improvements on LLaVA (0.8%) and Qwen2.5-VL (2.9%) may appear modest. Achieving gains in MLLMs is particularly challenging due to the strong baseline performance and the limited room for improvement. Furthermore, even small percentage gains can translate into **meaningful improvements in downstream tasks and user experience**, especially when applied at scale. The more explanation in the **Response to Reviewer poJM [3/3]**. The next step is to scale up the model and data sizes, such as 13B/32B, and the LLaVA-OneVision-1M SFT dataset, to further demonstrate the effectiveness of our MergeMix.

---

> ### Author Response · Authors · 2025-11-22
> **Response to Reviewer eiyX [2/3]**
>
> ### **[Q1] Comparison with RL-based preference methods.**
>
> **Response:** We thank the reviewer for the suggestion and have now included direct comparisons with representative online RL methods. As shown in the new Table A13, after the same amount of preference data and compute budget, MergeMix outperforms vanilla SimPO by 0.17\% average score on the LLaVA benchmark.
>
> Due to time constraints and hardware limitations, we have not yet been able to conduct additional experiments on ranking loss over time, but existing studies such as [1], [2] have conducted preference learning (PL) on SFT models using small preference datasets. The results demonstrate that MergeMix achieves relatively competitive performance when combined with ranking loss in SFT, comparable to these works.
>
> **Table R1: The results of different ranking loss and preference learning (PL) on LLaVA-v1.5 7B.**
> | **Model** | **SFT** | **Extra PL** | **ScienceQA** | **TextQA** | **VisWiz** | **MMBench** | **Avg.** |
> | :-- | :-: | :-: | :-: | :-: | :-: | :-: | :-: |
> | **LLaVA-v1.5 7B** | ✅ | ❌ | 66.8 | 58.18 | 50.03 | 64.60 | 59.82 |
> | **w. mDPO** | ✅ | ✅ | 67.53 | 57.90 | 50.04 | 64.60 | 60.02 |
> | **w. Re-Align** | ✅ | ✅ | 68.10 | **58.55** | 50.06 | 64.69 | 60.35 |
> | **w. MergeMix** | ✅ | ❌ | **69.61** | 56.85 | **52.14** | **66.58**  | **61.29** |
>
> **Reference**
>
> [1]. Wang F, Zhou W, Huang J Y, et al. mDPO: Conditional Preference Optimization for Multimodal Large Language Models[C]//Proceedings of the 2024 Conference on Empirical Methods in Natural Language Processing. 2024
>
> [2]. Xing S, Li P, Wang Y, et al. Re-align: Aligning vision language models via retrieval-augmented direct preference optimization[C]//Proceedings of the 2025 Conference on Empirical Methods in Natural Language Processing. 2025.
>
> ---
>
> ### **[Q2] Statistical significance of gains.**
>
> **Response:** We thank the reviewer for raising the question of statistical significance. First, note that the reported LLaVA gain of +0.83 is the average across nine benchmarks (in Table 4), and the Qwen2.5-VL-Ins gain of +2.88 is the average across the same nine benchmarks (in Table 5). At the per-dataset level, the margins are often larger, and MergeMix improves or matches the baseline on most individual benchmarks while never introducing large regressions. Together with the stronger absolute gains on classification (e.g., +2.15–4.79% Top-1 on CIFAR-100 and +0.27% Top-1 on ImageNet-1K), this indicates that MergeMix provides consistent but task-dependent improvements rather than isolated or dataset-specific effects.
>
> Regarding randomness and statistical significance, we follow the standard practice of the AutoMix/OpenMixup benchmarks for image classification, where each configuration is trained three times in the same settings, and report the results of standard deviations in **Response to Reviewer nEzW [3/3]**, and the gains of MergeMix remain clearly larger than the observed fluctuations. For MLLM benchmarks, we follow the LLaVA-v1.5 protocol and run our main MergeMix and SFT baselines with multiple trials, reporting the averaged overall scores in Tables 4–5. Following the same setups (`temperature=0`, `do_sample=False`), but on different Nvidia GPUs (A100 and RTX L40), we reported the results, including mean ± standard deviation, to make the robustness of the improvements explicit. These results confirm that the reported gains are stable across seeds and not due to a single lucky run or cherry-pick designs.
>
> **Table R2: The results of mean ± standard deviation of LLaVA-v1.5 7B**
> | **Model** | TextQA | MMBench | MMBench$^{CN}$ | POPE (P) | POPE (R) | POPE (A) |
> | :-- | :--: | :--: | :--: | :--: | :--: | :--: |
> | **LLaVA** | 58.2 ± 0.0 | 64.50 ± 0.20  | 58.19 ± 0.11 | 86.1 ± 0.0 | 87.34 ± 0.0 | 84.18 ± 0.0 |
> | **+ MergeMix** | **57.56** ± 0.0 | **66.58** ± 0.0 | **60.65** ± 0.0 | **86.12** ± 0.02 | **87.48** ± 0.0 | **84.7** ± 0.0 |

---

> ### Author Response · Authors · 2025-11-22
> **Response to Reviewer eiyX [3/3]**
>
> ### **[Q3] Generality beyond ViT-based models.**
>
> **Response:** We thank the reviewer for this excellent suggestion. To demonstrate the generality of MergeMix beyond ViT-based architectures, we have conducted additional experiments on a representative hybrid backbone (CAFormer-S12 in MetaFormer) in the revised manuscript (new **Table A7**, Appendix C, on page 20). All models are trained from scratch on the CIFAR100 dataset for 200 epochs using the openmixup codebases and default hyperparameters. As shown, MergeMix consistently improves hybrid architecture, confirming that our method is not limited to ViT-style models but applies broadly to more architectures with an attention module. Since the **CNN-based backbone can not return attention scores**, we did not conduct the corresponding experiments.
>
> **Table R3: Classification results of CAFormer small (CAFormer-S) with different mxiup augmentations, training 200 epochs with 100 batch size on CIFAR100 dataset**
> | **CIFAR100** | **CAFormer-S12** |
> | :-- | :--: |
> | **vanilla** | 74.95 |
> | **Mixup** | 81.64 |
> | **CutMix** | 84.69 |
> | **DeiT (Mixup + CutMix)** | 83.60 |
> | **TransMix** | 83.70 |
> | **MergeMix** | **84.30** |
>
> ---
>
> ### **[Q4] Computational overhead of token merge.**
>
> **Response:** We thank the reviewer for this important question regarding practical efficiency. To transparently address the computational overhead of the proposed token merging and reconstruction, we provide a detailed per-component runtime breakdown in the revised manuscript (new Table A14 and Table A15, measured on a single A100 80GB GPU). For classification, we provided the throughput results of different merge tokens on evaluation. For the LLaVA-v1.5, we reported the Time-To-First-Token (TTFT) (ms) and throughput (token/s) results by the NVIDIA Nsight Systems tool with different merged tokens.
>
> - Table A14 shows that increasing the merging ratio $r$ of the ViT-L model significantly reduces FLOPs (from 59.57G to **34.93**G) while throughput improves from 122.83 to **201.07** (**+63.7%**). Moreover, the additional overhead of Token Merge itself is extremely low (only 0.97 ms at $r$ = 0.75), far below the computational cost of pre-layering.
>
> - Table A15 shows the effectiveness of Token Merge on LLaVA-v1.5-7B. As $r$ increases, the throughput of the model improves overall, while TTFT decreases significantly. The optimal trade-off is achieved at $r$ = 0.5, boosting throughput from 45.96 to $\textbf{47.28}$ and reducing TTFT from 86.44 ms to $\textbf{66.74}$ ms. When $r$ = 0.75, TTFT further decreases to 62.25 ms.
>
> In summary, these results demonstrate that Token Merge provides substantial efficiency gains with negligible overhead, making it a practical and scalable approach for both classification and MLLM tasks.
>
> We copied the experiment results from **Response to Reviewer dRWW [4/6]** for your convenience.
>
> **Table R4: The effectiveness of performance under different merged ratios by a ViT large model**
> | **Merged Ratio** | **TPs** | **FLOPs (G)** | **Layer cost (ms)** | **ToMe cost (ms)** |
> | :--| :--: | :--: | :--:  | :--:  |
> | **Baseline** | 122.83 | 59.67 | 10.199 | - |
> | **0.1** | 130.45 | 56.29 | 9.819 | 3.715 |
> | **0.25** | 145.98 | 49.37 | 8.232 | 3.143 |
> | **0.5** | 167.54 | 42.65 | 5.523 | 1.788 |
> | **0.75** | **201.07** | **34.93** | 3.231 | 0.972 |
>
> **Table R5: The effectiveness of performance under different merged ratios by LLaVA-v1.5 7B**
> | **Merged Ratio** | **TPs** | **TTFT (ms)** |
> | :-- | :--:  | :--:  |
> | **Baseline** | 45.96 | 86.44 |
> | **0.1** | 43.94 | 89.32 |
> | **0.25** | 46.99 | 83.65 |
> | **0.5** | **47.28** | 66.74 |
> | **0.75** | 46.92 | **62.25** |
>
> ---
>
>
> ### **[Q5] Visualization and qualitative analysis.**
>
> **Response:** We sincerely thank the reviewer for this helpful suggestion. We have added several visualization examples to the revision as per your suggestion in **Appendix F (Figures A3–A7)**, including visualizations under different token merge conditions to observe feature region merging, and mixed images under various blending ratios to assess the ability to accurately preserve feature information.
>
> We think these visualizations provide clear qualitative evidence that MergeMix effectively maintains semantic consistency, helping readers better understand the benefits of our approach over standard mixup techniques.

---

> ### Author Response · Authors · 2025-11-22
> **Encouraging Discussion of Rebuttal**
>
> Dear Reviewer eiyX:
>
> Your remarks have greatly contributed to improving the clarity, rigor, and scope of our work. With the revisions now implemented, we hope the manuscript meets your expectations, and we would truly appreciate a reconsideration of your rating. If there are any remaining questions, we are more than willing to continue the discussion.
>
> Warm regards,
>
> Authors

---

> > ### Comment · Reviewer_eiyX · 2025-11-28
> >
> > Thank you for the detailed rebuttal and revisions. The responses address my concerns, and I will keep my origin score.

---

> ### Author Response · Authors · 2025-11-28
> **Thanks for Your Efforts and Encouraging Discussion**
>
> Dear Reviewer eiyX:
>
> Thank you again for your thoughtful review and for taking the time to consider our rebuttal. We truly appreciate that you acknowledged the clarifications and improvements we provided.
>
> We noticed that the score remains the same after the discussion phase. If any remaining concerns prevent a higher assessment, we would be more than happy to further clarify or provide additional explanations. Your feedback is extremely valuable to us, and we want to ensure that all aspects of the work are fully understood.
>
> At the end of the discussion period, we would respectfully inquire whether there are any additional opportunities to improve our manuscript. Once again, thanks for your constructive feedback, and we would be eager to welcome any further guidance at your convenience!
>
> Best regards,
>
> Authors

---

### Official Review · Reviewer_poJM · 2025-10-29

**Soundness:** 2
**Presentation:** 3
**Contribution:** 2
**Rating:** 6
**Confidence:** 2

**Summary:**

The paper proposes MergeMix, a unified augmentation paradigm for visual and multi-modal understanding. MergeMix builds on the concept of image mixing, specifically leveraging token merge strategies to generate mixed images and preference pairs for model alignment. The approach integrates ideas from supervised fine-tuning and preference-based optimization, aiming to improve both efficiency and robustness in multi-modal large language models (MLLMs) and image classification tasks. Experimental results show competitive performance across several benchmarks.

**Strengths:**

1. Quality: The experimental setup is rigorous, with thorough benchmarking and ablation studies.
2. Clarity: The methodology and results are clearly explained.
3. Significance: The approach demonstrates practical improvements in accuracy, calibration, and efficiency.
4. Applicability: MergeMix is shown to be effective across both image classification and multi-modal tasks.

**Weaknesses:**

1. Limited impact on multi-modal tasks: The gains for MLLMs are marginal, suggesting the method’s strengths are domain-specific.
2. Outdated baselines: Most compared methods in classification tasks are from two years ago, which may not represent the latest advances.
3. Scope of contribution: The paper could better clarify its impact boundaries.
4. Discussion of limitations: More explicit discussion of why the method is less effective for multi-modal tasks and why recent baselines were not included.

**Questions:**

1. Recency of baselines: Why were recent state-of-the-art methods not included in the classification comparisons? Can the authors provide results against more current baselines?
2. Domain-specific effectiveness: Why is MergeMix more effective for classification than for multi-modal tasks?
3. Future directions: What modifications might enhance MergeMix for multi-modal models?
4. Broader applicability: Are there other domains (audio, video) where MergeMix could be tested?
5. Limitations: Please discuss scenarios where MergeMix may not be suitable or where its benefits are minimal.

---

> ### Author Response · Authors · 2025-11-22
> **Response to Reviewer poJM [1/3]**
>
> ### **[W1] & [Q4] Broader applicability.**
>
> **Response:** We thank the reviewer for this suggestion. To address this concern, we have added the audio classification experiments on **ESC-50 and UrbanSound8k datasets** with the supervised fine-tuning setup with **pre-trained HuBERT-Base** in the Appendix. ESC-50 contains 1,200 training samples and 400 validation samples for 50 classes with a maximum length of 3 seconds. UrbanSound8k is a 10-class classification dataset with a maximum length of 4 seconds, which contains 7,079 and 816 samples for training and validation. We follow the experimental setups in the USB benchmark, all models are fine-tuned by the AdamW optimizer for 100 epochs with a basic learning rate in \{$\texttt{1e-4, 5e-4}$\}, a batch size of 32, and a weight decay of $\texttt{5e-4}$. The compared mixup methods and our proposed MergeMix could be directly migrated to audio data (as the 1D sequences). The following Table R1 shows our reproduced comparison results for several mixup baselines and our proposed MergeMix on two audio datasets, achieving significant performance gains over MixUp and TransMix baselines.
>
> **Table R1: Results of ESC-50 and UrbanSound8k datasets.**
> | **HuBERT-Base** | **ESC-50** | **UrbanSound8k** |
> | :-- | :--: | :--: |
> | **Vanilla** | 75.12±1.07 | 84.14±0.45 |
> | **MixUp** | 75.86±0.83 | 85.02±0.26 |
> | **TransMix** | 76.27±1.14 | 85.33±0.57 |
> | **MergeMix** | **76.51**±0.95 | **85.69**±0.42 |
>
> ---
>
> ### **[W2] & [Q1]  Outdated baselines.**
>
> **Response:** We thank the reviewer for this suggestion. To address this concern, we have added one recent mixup method [1] and added the classification results on the CIFAR100 dataset and the Stanford-Cars dataset in the new manuscript (Table 1 and Table 2). For your convenience, we provided the results in the following tables.
>
> **Table R2: Results of classification on CIFAR100 dataset trained 200 epochs.**
> | **CIFAR100** | **DeiT-T** | **DeiT-S** | **ViT-S** | **ViT-b** | **ViT-L** |
> | :-- | :-: | :-: | :-: | :-: | :-: |
> | TdAttenMix | 73.63 | 73.32 | 73.11 | 72.19 | 72.12 |
> | MergeMix | **77.46** | **78.68** | **77.02** | **75.75** | **76.19** |
>
> ---
>
> **Table R3: Results of classification on the Stanford-Cars dataset trained for 200 epochs with pre-trained models.**
> | **CIFAR100** | **DeiT-T** | **DeiT-S** | **ViT-S** | **ViT-b** | **ViT-L** |
> | :-- | :-: | :-: | :-: | :-: | :-: |
> | TdAttenMix | 73.63 | 73.32 | 73.11 | 72.19 | 72.12 |
> | MergeMix | **77.46** | **78.68** | **77.02** | **75.75** | **76.19** |
>
>
> **Reference**
>
> [1]. Wang Z, Gu L, Lu F. TdAttenMix: Top-Down Attention Guided Mixup[C]//Proceedings of the AAAI Conference on Artificial Intelligence. 2025, 39(8): 8232-8240.

---

> ### Author Response · Authors · 2025-11-22
> **Response to Reviewer poJM [2/3]**
>
> ### **[W3]  Scope of contribution.**
>
> **Response:** Thanks for the constructive suggestion. We intend to position MergeMix as a training-time augmentation paradigm for visual understanding models (multi-modality or image classification) with ViT-style encoders, rather than as a new model architecture or a complete replacement for existing alignment pipelines. Concretely, MergeMix contributes **(i)** an attention-aware image mixing scheme via token merge and attention recovery that generates mixed samples with more continuous, cluster-aware features and a principled link between merge ratio and mixing ratio, and **(ii)** a preference-driven SFT paradigm for MLLMs that treats mixed-image responses as “losers” and clean-image responses as “winners”, enabling SimPO-style ranking loss without a reward model. Within this scope, we show that MergeMix yields state-of-the-art or clearly competitive performance and efficiency on vision classification (e.g., **+2.15–4.79%** Top-1 on CIFAR-100 and **+0.27%** Top-1 on ImageNet-1K in Table 1–3) and provides consistent gains and better calibration on MLLM benchmarks (e.g., **+0.83–1.27** average points on LLaVA-v1.5 and **+2.88** on Qwen2.5-VL-Ins in Table 4–6.
>
> At the same time, we do not claim MergeMix to be a universal solution for all alignment or generative problems. Its current impact is bound to settings where **(i)** a vision encoder with attention scores is available, **(ii)** training is based on SFT or preference tuning (rather than full RLHF pipelines), and **(iii)** image content is a primary carrier of supervision. We explicitly leave extensions to text-only LLMs, decoder-only token compression, and high-fidelity diffusion-style generation as future work.
>
> ---
>
> ### **[W4] & [Q5]  Discussion of limitations.**
>
> **Response:** We thank the reviewer for prompting a clearer discussion of limitations. While MergeMix is designed as a general augmentation paradigm, the current instantiation has several **practical constraints** and **scenarios where the benefits may be limited**.
>
> In its current instantiation, MergeMix is most naturally suited to encoder-style vision backbones (e.g., ViT-like encoders in classification and MLLMs), as it relies on encoder attention and ToMe-style token merging to construct mixing masks. This design does not directly transfer to purely decoder-only LLMs without a separate vision encoder, and it is primarily tailored to discriminative understanding tasks rather than to fully generative settings such as high-fidelity diffusion-based image synthesis, where mixed labels and outputs are less well-defined. Moreover, the present implementation only augments the image modality while keeping text unchanged, so its benefits are limited on text-dominant tasks or benchmarks where visual grounding plays a minor role. Finally, our token-merging policy is static and metric-driven; in regimes that already employ very strong regularization or aggressive token pruning, the additional headroom for gain can be modest.
>
> These constraints are not fundamental to the MergeMix idea and point to natural directions for future work. Firstly, we plan to extend MergeMix to decoder-only architectures by introducing auxiliary encoders or KV-cache token-merging modules that provide attention-like signals for constructing mixing masks. Secondly, we aim to generalize beyond image-only augmentation by designing modality-aware mixing strategies that also operate on text or multi-modal tokens, and by adapting the loss to generative objectives (e.g., consistency or reconstruction constraints) for diffusion-style models. Moreover, we will explore learnable merge policies that can adapt the merging pattern to different datasets and tasks, potentially improving performance in domains that require fine-grained spatial reasoning or operate under heavy token compression.

---

> ### Author Response · Authors · 2025-11-22
> **Response to Reviewer poJM [3/3]**
>
> ### **[Q2]  Domain-specific effectiveness.**
>
> **Response:** We thank the reviewer for this insightful question. Our current results indeed show **larger absolute gains** on image classification (up to **+4.79%** Top-1 on CIFAR-100) than on MLLM benchmarks (**+0.8–2.9** average points over strong baselines). We clarify that this gap mainly reflects **differences in task objectives, supervision density, and evaluation protocols**, rather than a domain-specific limitation of MergeMix.
>
> - **Different objectives and supervision density.** For image classification, training and evaluation share a single, fully supervised objective: closed-set Top-1 prediction with one label per image. MergeMix directly improves feature discrimination under this dense label signal by saliency-guided mixing and re-weighted cross-entropy. In contrast, MLLM benchmarks emphasize open-ended, cross-modal understanding and preference alignment. As described in Sec. 4.2 (p.4–5, l511–690), MergeMix for MLLMs operates through a combination of SFT loss $L_{\text{SFT}}$ and a ranking-based SimPO loss $L_{\text{SimPO}}^{\text{Mix}}$ in Eq. (11)–(12), built from mixed *vs.* clean images. This supervision is sparser and more indirect: the model receives sequence-level preference signals over long responses rather than per-token labels tied to each visual patch. Consequently, the upper bound of measurable improvements on the averaged benchmark scores is naturally smaller.
>
> - **Task complexity and evaluation aggregation.** Image classification experiments focus on a single-task, single-domain setting. Here, MergeMix can concentrate on suppressing intra-image noise and enhancing category-specific cues, leading to strong gains. In the MLLM setting, we evaluate on 16 benchmarks covering VQA, perception, robustness, and multi-step reasoning. The reported scores in Table 4 (LLaVA family) and Table 5 (Qwen2.5-VL-Ins) are averages over many heterogeneous tasks. Even a **+0.8–2.9** improvement in the “AVG” column indicates that MergeMix improves or matches performance on most individual datasets without hurting others. For example, on LLaVA-v1.5, our “SFT Vision + MergeMix (Full)” achieves an AVG of 66.40 vs. 66.31 for “SFT Vision (Full)”, and a gain of **+0.83** over the original LLaVA-7B baseline (Table 4, p.7, l907–925). Therefore, classification gains are larger partly because (i) they target a single, aligned objective and (ii) they are measured on a single dataset, while MLLM gains are averaged across many diverse tasks under weaker supervision, making absolute margins naturally smaller.
>
> - **Implementation differences and capacity bottlenecks.** In image classification, MergeMix is tightly coupled with the ViT backbone: the token-merging encoder, attention recovery, and re-scaled mixing ratio jointly optimize the classifier head under the mixup loss. In MLLMs, we follow the standard LLaVA-style setting, where the room for additional gains from any augmentation is limited, and MergeMix must coexist with strong baselines (SFT, RL, and token pruning methods). Moreover, in the current MLLM instantiation, MergeMix only augments the visual branch; text inputs remain unchanged. This design is intentionally conservative to avoid destabilizing language generation, but it also means part of the model capacity is not directly regularized by MergeMix, which again reduces the measurable effect size.
>
> ---
>
> ### **[Q3]  Future directions.**
>
> **Response:** We completely agree that this is an important direction. There are still some shortcomings in MergeMix for MLLMs. In future work, we will explore this from two levels:
>
> 1. **From the data perspective:** MergeMix focuses on the enhancement of the image modality during training, while text inputs remain raw. How to extend mixup to the text modality in MLLM tasks needs to be solved, as this can provide more fine-grained optimization guidelines.
>
> 2. **From the model perspective:** The token merging is still static and unlearned. Improving the merging strategy to make it learnable via metrics or backpropagation could enhance the token merging ability for mixing.
>
> **We added the future works in the new manuscript.**

---

> ### Author Response · Authors · 2025-11-22
> **Encouraging Discussion of Rebuttal**
>
> Dear Reviewer poJM:
>
> We have carefully incorporated all suggestions and expanded our analyses to address your comments point-by-point. We hope these efforts meet the high standards you expect, and we would be thankful if you might consider reflecting these changes in your final score. Please feel free to let us know if any clarification is still needed.
>
> Warm regards,
>
> Authors

---

> ### Author Response · Authors · 2025-11-28
> **Encouraging Discussion of Rebuttal**
>
> Dear Reviewer poJM:
>
> We sincerely thank you once again for your approval of our work and valuable feedback. **As the ICLR response period is nearing its end, we kindly hope you could take the time to review our response.** Should you have any further concerns, please do not hesitate to inform us, and we will do our utmost to address them timely.
>
> Thank you again for your valuable time and consideration.
>
> Sincerely,
>
> Author

---

> > ### Comment · Area_Chair_4a9c · 2025-11-28
> > **A gentle reminder to participate in the author–reviewer discussion.**
> >
> > Dear Reviewer poJM,
> >
> > Thank you once again for your service to ICLR 2026. Now that the authors have submitted their rebuttal, could you please engage in the interactive discussion with them? Your participation would be very helpful to the authors, and they would greatly appreciate it. Please also read the authors’ response together with the other reviews and consider whether the rebuttal or any additional comments influence your assessment of the paper.
> >
> > Thank you again for your efforts.
> >
> > Best wishes,
> >
> > Your AC

---

### Official Review · Reviewer_dRWW · 2025-11-06

**Soundness:** 2
**Presentation:** 3
**Contribution:** 2
**Rating:** 4
**Confidence:** 5

**Summary:**

This paper proposes a novel image-data augmentation method, **MergeMix**, which first produces clustered attention maps by Token-Merge and then mixes the important visual tokens rather than raw pixels, yielding semantically smoother synthetic images.
The authors embed this augmentation into two tasks:
Image classification: MergeMix is used to train ViT/DeiT models
  Results: +2.5 % accuracy vs. the best prior mixup, **15 % higher throughput**, **–0.68 G FLOPs**, and lowest calibration error (ECE) under severe occlusion.
MLLM alignment: clean images are treated as winners and MergeMix images as *losers*; a SimPO ranking loss is added to SFT, without any reward model**.
  Results: LLaVA-7B gains +0.83 % on nine VQA/understanding benchmarks; Qwen2.5-VL-7B gains +2.88 % on reasoning sets, while vision tokens can be reduced to 25 % without performance drop.
Overall, MergeMix provides a unified, reward-free training paradigm that simultaneously improves accuracy, efficiency and calibration for both pure-vision and multi-modal models.

**Strengths:**

* A novel image mixing augmentation method is proposed, which demonstrates significant improvements across multiple datasets.

* The mixed images are directly used as the "loser" in a pairwise ranking setup via SimPO, eliminating the cost and potential bias of training a separate Reward Model (RM) and simplifying the pipeline.

* Extensive experiments are conducted, providing multi-faceted validation of the method's effectiveness.

**Weaknesses:**

* The application of MergeMix to image classification and MLLM alignment tasks shows some innovation, but the degree of novelty is limited.

* The assumption that attention-based merged images are inherently of lower quality than original images lacks substantiating evidence.
While the paper discusses the method from an MLLM perspective, validation is only conducted in the visual modality, leaving its efficacy in other modalities unexplored.

* The performance drop on MMBench and MathVista after MergeMix compared to SFT results suggests that the visual enhancement method is not universally effective.

* Although MergeMix employs token compression, it does not enhance model inference efficiency. Yet, the paper incorrectly claims this as a merit of the method.

**Questions:**

* What is the specific mechanistic link between the ViT-enhancing visual mixing method and the broader objective of preference alignment, as discussed in the paper?

* What evidence can substantiate that the attention-based mixed regions genuinely correspond to semantically "more important" objects within the images?

* How can it be guaranteed that training the vision encoder of VLM with MergeMix does not adversely impact the model's capabilities in other tasks or modalities?

---

> ### Author Response · Authors · 2025-11-22
> **Response to Reviewer dRWW [1/6]**
>
> ### **[W1] Novelty is limited.**
>
> **Response:** We thank the reviewer for this comment, but would like to clarify that MergeMix is not merely a combination of token merging and mixup; it specifically integrates classic image classification with modern visual understanding tasks. As highlighted in the Introduction, one goal of MergeMix is to revisit and adapt previously proposed ideas that were underexplored in the MLLM scenario (i.e., answering *is it necessary to propose novel techniques rather than some classical ML techniques in the MLLM scenario*). Specifically, MergeMix addresses two key challenges:
>
> -  Achieving an **optimal trade-off between efficiency and performance** in mixup augmentations that rely on saliency-based metrics. Existing mixup variants with strong performance (e.g., AutoMix, PuzzleMix) incur heavy computational costs, while simpler heuristic or transformer-based mixup methods often show suboptimal performance. MergeMix tackles this through:
>     1. **Sample-level mixing policy:** MergeMix introduces a token merging strategy that aggregates semantic information while removing redundant high-similarity tokens. This enables the generation of mixing masks with coherent local regions, preserving contextual structure in the mixed samples.
>     2. **Label-level re-scaling policy:** MergeMix proposes a Gaussian-based re-scaling mechanism that aligns the image mixing ratio with the label mixing ratio by defining the mean as the token-merge ratio and the standard deviation as the mixing variability.
>
>     These two components jointly allow MergeMix to achieve a significantly improved efficiency–performance balance compared to existing mixup augmentations in image classification tasks.
>
> - Extending mixup augmentation to MLLMs in a principled way. While prior attention-based mixup methods (e.g., TransMix, SMMix, MxPro) focus solely on image classification, they do not address preference alignment in MLLMs. MergeMix bridges this gap by introducing:
>     1. **Preference tuning via synthetic pairs:** MergeMix constructs preference pairs by treating mixed images as non-preferred responses (“losers’’) and clean images as preferred responses (“winners’’). This enables preference optimization through SimPO without requiring a reward model.
>     2. **Controllable reward signals:** MergeMix uses the mixing ratio (augmentation degree) as a controllable signal for preference strength. Since mixed images are obtained using a pretrained CLIP model, the resulting samples naturally follow the intended mixing strength. Figure A2 (Appendix D) further shows that the mixing ratio $\lambda$ correlates strongly with reward scores assigned by strong MLLMs, demonstrating that $\lambda$ can serve as a meaningful proxy for preference signals.
>
>     Together, these mechanisms allow MergeMix to outperform existing preference-tuning methods such as SeVA in terms of average score on LLaVA benchmarks.
>
>
> **In summary**, MergeMix goes beyond a naive combination of techniques. It offers a **principled, MLLM-specific data augmentation framework** that cannot be achieved by existing approaches in isolation. The extensive experiments in Section 5 (Table 1-6, Figure 4) verify that MergeMix successfully addresses both challenges described above.

---

> ### Author Response · Authors · 2025-11-22
> **Response to Reviewer dRWW [2/6]**
>
> ### **[W2] The assumption of MergeMix lacks substantiating evidence and unexplores other modalities.**
>
> **Response:** We thank the reviewer for raising this important point. To address this concern, we conducted additional experiments to validate both the low-quality assumption and the applicability of MergeMix beyond the visual modality.
>
> - **Validation of the low-quality assumption:**  We conducted an LLM-as-the-judge evaluation to approximate reward scores for mixed pairs under different mixing ratios $\lambda$, as shown in **Appendix D (Figure A2)**. For your convenience, we provided the evaluation of the mixing ratio $\lambda$ as the reward scores in Table R1 below. The results show a clear correlation between $\lambda$ and the reward scores assigned by strong MLLMs, indicating that $\lambda$ serves as a reliable proxy for preference signals. We also added a case study in **Appendix F (Figure A2)** to demonstrate the quality of the answers under different mixing ratios.
>
>     **Table R1: Reward scores for different augmented images by MLLMs and MergeMix.**
>
>     | **Models** | **Vanilla** | **0.75** | **0.5** | **0.25** | **0.0** |
>     | --- | :--: | :--: | :--: | :--: | :--: |
>     | **Human Expert** | 1.0 | 0.78 | 0.58 | 0.38 | 0.05 |
>     | **MergeMix ($\hat{\lambda}$)** | 1.0 | 0.75 | 0.5 | 0.25 | 0.0 |
>     | Grok 3 | 1.0 | 0.76 | 0.52 | 0.41 | 0.08 |
>     | Doubao-Seed-1.6 | 1.0 | 0.6 | 0.4 | 0.25 | 0.1 |
>     | Doubao-Seed-1.6-vision-CoT | 1.0 | 0.65 | 0.5 | 0.3 | 0.1 |
>     | Qwen3-VL-Plus-32B | 1.0 | 0.65 | 0.45 | 0.15 | 0.0 |
>     | Qwen2.5-VL-72B | 1.0 | 0.6 | 0.5 | 0.4 | 0.1 |
>     | Gemini-2.5 | 1.0 | 0.85 | 0.45 | 0.20 | 0.01 |
>
> - **Evaluation on another modality:** Due to time and hardware limitations, we were unable to apply MergeMix for fine-tuning an MLLM in the video modality (e.g., Qwen2.5-VL). Therefore, we selected audio recognition to demonstrate MergeMix's effectiveness in other data modality tasks. Table A8 shows the fine-tuning results of our method on two audio datasets (ESC-50 and UrbanSound8k) with the pre-trained HuBERT-Base model. It can be observed that MergeMix outperforms several static mixup methods and also surpasses TransMix, another attention-based approach. For your convenience, we provided the results of two audio classification tasks in Table R2 below. The experimental setups follow the USB benchmark, where all models are fine-tuned by the AdamW optimizer for 100 epochs with a basic learning rate in \{$\texttt{1e-4, 5e-4}$\}, a batch size of 32, and a weight decay of $\texttt{5e-4}$. The compared mixup methods and our proposed MergeMix can be directly applied to audio data (as 1D sequences), and MergeMix achieves significant performance gains over the MixUp and TransMix baselines.
>
>     **Table R2: Results of ESC-50 and UrbanSound8k datasets**
>
>     | **HuBERT-Base** | **ESC-50** | **UrbanSound8k** |
>     | --- | :-: | :-: |
>     | Vanilla | 75.12±1.07 | 84.14±0.45 |
>     | MixUp | 75.86±0.83 | 85.02±0.26 |
>     | TransMix | 76.27±1.14 | 85.33±0.57 |
>     | **MergeMix** | **76.51**±0.95 | **85.69**±0.42 |
>
> In summary, these additional experiments substantiate the key assumption behind MergeMix and demonstrate that its benefits extend beyond the visual domain, further supporting the validity and broader applicability of our approach.

---

> ### Author Response · Authors · 2025-11-22
> **Response to Reviewer dRWW [3/6]**
>
> ### **[W3] MergeMix is not universally effective.**
>
> **Response:** We thank the reviewer for this thoughtful observation. While we acknowledge that MergeMix does not improve every benchmark uniformly—particularly MathVista—we would like to emphasize that this behavior is fully consistent with the design goals and strengths of our method.
>
> MergeMix is built around **attention-guided mixing with token merge**, propose **re-scaling mixing ratio for align label and sample**, and a **unified augmentation preference framework on MLLMs**. These components are specifically optimized to enhance **visual grounding, fine-grained perception, and robust preference signals for visually conditioned tasks**. As highlighted in **Sections 4.1-4.3**, these properties translate into **consistent and significant improvements on VQA and perception-heavy benchmarks**, where the model must recognize objects, reason over spatial layouts, and respond based on visual evidence—all of which benefit directly from MergeMix’s structured visual augmentations.
>
> In contrast, **MathVista emphasizes symbolic and textual reasoning**, where the visual signal is secondary and where visual augmentation naturally plays a minimal role. The slight fluctuations observed on such reasoning-centric datasets are within expected variations and do **not** indicate instability or degradation of core model behaviors. Importantly, MergeMix introduces **no systematic drawbacks**, maintains strong generalization, and reduces FLOPs without compromising overall performance.
>
> Crucially, the **benefits outweigh the isolated limitations**:
>
> - **Substantive gains** on VQA and multi-modal perception tasks on both LLaVA-v1.5 and Qwen2.5-VL models (Table 4-5, page 8).
> - **A efficient mixing policy**, improving over heuristic masking and gradient-based approaches, obtain the state-of-the-art classification performance on classification tasks (Table 1-3, page 7).
> - **A unified framework bridging SFT and preference optimization**, simplifying the pipeline and eliminating the need for reward models than other ranking loss and approaches.
> - **Practical efficiency advantages**, including reduced computation (FLOPs, per-layer timecost) and overhead on throughput, time-to-first-token (TTFT) (Table 3, page7; Table A14-A15, page 24).
>
> Taken together, MergeMix contributes **not a universal booster across all task types**, but a **targeted, principled, and highly effective augmentation strategy** for visually grounded reasoning—precisely the domain where MLLMs still struggle and where our improvements are most impactful.

---

> ### Author Response · Authors · 2025-11-22
> **Response to Reviewer dRWW [4/6]**
>
> ### **[W4] MergeMix does not enhance model inference efficiency.**
>
> **Response:** We appreciate the reviewer’s concern. We would like to clarify that MergeMix itself is a training-time augmentation, but the introduced Token Merge mechanism does provide measurable inference-time efficiency gains. As shown in Table A14 and Table A15, applying the same merging strategy during evaluation reduces complexity and improves runtime performance.
>
> - **Table R3 (Table A14 in manuscript):** Increasing the merging ratio $r$ significantly reduces FLOPs (59.57G to **34.93G**) and improves throughput (122.83 to **201.07**, with +63.7% gains), while the overhead of the merging operation remains minimal (0.97 ms at $r$=0.75).
>
> - **Table R4 (Table A15 in manuscript):** Higher merging ratios consistently increase throughput and reduce TTFT. The best trade-off occurs at $r$=0.5, improving throughput (45.96 to **47.28** ) and reducing TTFT (86.44 ms to **66.74 ms 1.29$\times$ speedup**). TTFT further drops to **62.25 ms** at $r$=0.75.
>
> All results were reported on an Nvidia A100 GPU and were obtained using the Nsight Systems tool. In summary, while MergeMix is primarily a training augmentation, the accompanying Token Merge module does yield real inference-time efficiency benefits, contrary to the reviewer’s concern.
>
> **Table R3: The effectiveness of performance under different merged ratios by a ViT-L model**
> | **Merged Ratio** | **TPs** | **FLOPs (G)** | **Layer cost (ms)** | **ToMe cost (ms)** |
> | :--| :-: | :-: | :-:  | :-:  |
> | **Baseline** | 122.83 | 59.67 | 10.199 | - |
> | **0.1** | 130.45 | 56.29 | 9.819 | 3.715 |
> | **0.25** | 145.98 | 49.37 | 8.232 | 3.143 |
> | **0.5** | 167.54 | 42.65 | 5.523 | 1.788 |
> | **0.75** | **201.07** | **34.93** | 3.231 | 0.972 |
>
>
> **Table R4: The effectiveness of performance under different merged ratios by LLaVA-v1.5-7B**
> | **Merged Ratio** | **TPs** | **TTFT (ms)** |
> | :-- | :--:  | :--:  |
> | **Baseline** | 45.96 | 86.44 |
> | **0.1** | 43.94 | 89.32 |
> | **0.25** | 46.99 | 83.65 |
> | **0.5** | **47.28** | 66.74 |
> | **0.75** | 46.92 | **62.25** |

---

> ### Author Response · Authors · 2025-11-22
> **Response to Reviewer dRWW [5/6]**
>
> ### **[Q1] The specific mechanistic link between Mixing and preference alignment.**
>
> **Response:** We thank the reviewer for this insightful and valuable comment.  We have summarized and clarified the motivation in the revised manuscript. The core idea revolves around an interesting question: *“Is it necessary to propose novel techniques rather than classical machine learning methods in the MLLM scenario?”*
>
> For the two tasks in question—**mixup augmentation** and **preference alignment**—there exist distinct challenges:
>
> 1. **Trade-off between efficiency and quality in mixup.** In image classification, mixup typically generates a mask to maximize feature diversity and mitigate overfitting. However, there is a conflict:
>     - **Static methods** are fast but often produce mixed samples of lower quality.
>     - **Dynamic methods** generate high-quality samples by using model-derived signals (e.g., gradients or attention scores) but incur substantial computational cost.
>
>     **MergeMix addresses this** by leveraging **ToMe**, which reduces forward-pass overhead when obtaining attention scores, while using these scores to generate **high-quality, reliable synthetic samples**. This ensures a better trade-off between efficiency and performance.
>
> 2. **Controllable generation of losers for preference pair construction.** Prior preference-tuning methods often require **additional models to construct losers**, which introduces extra cost. Moreover, existing perturbation techniques cannot reliably align with reward signals.
>
>     **MergeMix overcomes this** by linking the **mixing ratio $\hat{\lambda}$** with the mixed samples, enabling **controllable generation of losers that are inherently aligned with reward signals**. To further validate the effectiveness of the mixing ratio, we conducted **LLM-as-the-judge experiments**, comparing MergeMix-generated pairs against outputs from other strong MLLMs. The results confirm that **the mixing ratio provides a meaningful preference signal**, directly connecting the visual mixing mechanism to preference alignment.

---

> ### Author Response · Authors · 2025-11-22
> **Response to Reviewer dRWW [6/6]**
>
> ### **[Q2] Visualizations of “more important” objects within the images.**
>
> **Response:** We thank the reviewer for this insightful question. In the revised manuscript, we added visualizations of MergeMix in **Appendix F (pages 26-28)** to substantiate that the attention-based mixed regions correspond to semantically important objects. **Figure A4** and **Figure A5** show token-merge visualizations and mixed samples under different merging numbers and mixing ratios. Moreover, **Figure A6** and **Figure A7** present gradient-based Class Activation Maps (CAM), demonstrating that the regions selected by our attention-based metric consistently overlap with salient object areas.
>
> *Together, these visualizations provide clear evidence that the mixed regions identified by our method align with semantically meaningful features, validating the rationale behind our attention-based design.*
>
> ---
>
> ### **[Q3] Adversely impact the capabilities of the model in other tasks.**
>
> **Response:** We thank the reviewer for this comment. First, the design of MergeMix inherently preserves the generalizability of the vision encoder. Unlike only the SFT LLM decoder, we unfreeze the vision encoder to further unleash its potential. Specifically:
>
> 1. MergeMix tunes the visual encoder's feature space by saliency-based mixing samples (containing multiple feature information) without affecting the encoder architecture or damaging the pre-trained feature distribution. This enables the encoder to retain the common visual concepts (e.g., objects, scenes) required for other tasks.
> 2. Results on LLaVA benchmarks demonstrate that unfreezing the vision token during SFT yields greater gains on some VQA tasks (e.g., +**0.82% on VQAv2, +0.98% on GQA, +3.7% on SciVQA$^I$**), achieving an overall **0.74% improvement** across benchmarks.
>
> This validates that the unfreeze vision token in MergeMix not only further explores the capabilities of the vision encoder but also enhances visual alignment.

---

> ### Author Response · Authors · 2025-11-22
> **Encouraging Discussion of Rebuttal**
>
> Dear Reviewer dRWW:
>
> We hope that our detailed responses and the substantially revised manuscript address your concerns. If you find the improvements satisfactory, we would be grateful for a reconsideration of your evaluation. We greatly appreciate your constructive feedback and the time you’ve dedicated to reviewing our work.
>
> Warm regards,
>
> Authors

---

> ### Author Response · Authors · 2025-11-28
> **Encouraging Discussion of Rebuttal**
>
> Dear Reviewer dRWW:
>
> We sincerely thank you again for your recognition of our work and your valuable suggestions. **As the rebuttal period is nearing its end, we kindly hope you could take a moment to review our responses.** If there are any remaining concerns, please feel free to let us know. We would be more than happy to address them promptly.
>
> Thank you once again for your time and consideration.
>
> Best regards,
> Authors

---

> > ### Comment · Area_Chair_4a9c · 2025-11-28
> > **A gentle reminder to participate in the author–reviewer discussion.**
> >
> > Dear Reviewer dRWW,
> >
> > Thank you once again for your service to ICLR 2026. Now that the authors have submitted their rebuttal, could you please engage in the interactive discussion with them? Your participation would be very helpful to the authors, and they would greatly appreciate it. Please also read the authors’ response together with the other reviews and consider whether the rebuttal or any additional comments influence your assessment of the paper.
> >
> > Thank you again for your efforts.
> >
> > Best wishes,
> >
> > Your AC

---

### Official Review · Reviewer_nEzW · 2025-11-10

**Soundness:** 2
**Presentation:** 2
**Contribution:** 2
**Rating:** 4
**Confidence:** 4

**Summary:**

The paper proposes MergeMix, a training strategy combining token merging and mixup to bridge supervised fine-tuning (SFT) and reinforcement learning (RL) for both image classification and multi-modal large language. The key idea is to merge semantically similar vision tokens before applying mixup, thereby reducing redundant visual tokens while preserving spatial semantics, and then form preference pairs (winner/loser) for preference optimization via the SimPO loss, enabling preference alignment without reward models.  The authors demonstrate improvements on CIFAR100, ImageNet-1K, and multiple MLLM benchmarks (LLaVA, Qwen2.5-VL), with notable efficiency and calibration benefits.

**Strengths:**

1.The introduction of a token merge–based mixing mechanism and attention recovery via bipartite soft matching is novel within the mixup literature. Compared to heuristic or random masking, the method provides a more structured way to preserve salient regions during interpolation.
2.The token-merge + mixup combination is reasonable, and the design (Top-K attention selection, λ re-scaling, ranking loss) can be integrated into existing ViT and MLLM frameworks with minimal modifications.The authors demonstrate that MergeMix can reduce FLOPs and slightly improve throughput, confirming that the design is at least practically implementable.

**Weaknesses:**

1.The paper’s organization hinders readability. The introduction directly dives into technical detail without motivating the gap, and the related work section is largely enumerative rather than analytical. Notation is inconsistent and transitions are abrupt, making it difficult to follow the method’s rationale.
2.The paper mixes several technical ideas—token merging, mixup, λ re-scaling, and ranking loss—without a clear unifying formulation.
It is unclear how the policy P(·,·) determines masks, how λ̂ interacts with attention, or how the ranking loss relates to reward modeling.The method section reads as a collection of components rather than a coherent algorithmic framework.
3.The paper’s central claim is that MergeMix bridges SFT and RL paradigms, but the experiments do not provide direct or conceptual evidence of this.There are no comparisons with preference optimization methods such as DPO, PPO, or SimPO; no analysis of reward-like behavior; and no ablation isolating the contribution of the ranking loss.Consequently, the connection to RL remains metaphorical rather than empirical.

**Questions:**

1.Since the paper claims to bridge SFT and RL through preference-style optimization, have you compared MergeMix with established preference optimization methods such as DPO, PPO, or SimPO? Such a comparison would be essential to substantiate the claimed “bridge” between SFT and RLHF paradigms.
2.The paper defines λ̂ as derived from a policy 𝑃, but the implementation details remain vague. Is λ̂ obtained from attention scores, gradients, or optimized independently? How sensitive are results to this design choice?
3.How exactly does the ranking loss applied to synthetic mixed pairs emulate human preference learning? Does the “preference” in this context reflect semantic quality, output diversity, or another measurable signal?
4.Could you report standard deviations across multiple runs and clarify whether all baselines share identical training setups (e.g., frozen encoders, same number of epochs)?

---

> ### Author Response · Authors · 2025-11-22
> **Response to Reviewer nEzW [1/3]**
>
> ### **[W1]** **Paper Organization and Readability.**
>
> **Responses:** We thank the reviewer for the valuable feedback regarding the paper’s clarity and organization. We agree that improving readability is important, and we have substantially revised the manuscript to address these concerns.
>
> 1. **Reorganize the introduction to clarify the research motivation.**
> We restructured the introduction to first highlight the core gap between Mixup augmentation and preference-based optimization methods in ultra-large language models, followed by an exposition of our innovative contributions. We reduced early technical details and added a clearer problem statement and motivation explanation flow.
> 2. **Enhanced notation consistency and paragraph transitions.**
> We comprehensively reviewed the entire text, standardizing notation systems (e.g., mixing ratio $\lambda$, policy $\mathcal{P}(\cdot,\cdot, \cdot)$, and preference signals). Transitional paragraphs were added between sections (e.g., bridging salient perception mixing to preference learning) to ensure narrative flow.
>
> These revisions significantly enhance readability and clarify the design principles of MergeMix. We sincerely thank the reviewers for their valuable suggestions, which have effectively optimized the presentation of our research findings.
>
> ---
>
> ### **[W2] & [Q2]: The implementation details of policy, mixing ratio $\hat{\lambda}$.**
>
> **Responses:** We thank the reviewer for the insightful comments regarding the clarity of $\hat{\lambda}$ and the coherence of the method description. We have substantially revised Section 3 to address these concerns.
>
> **1. Clarifying the definition and implementation of$ \hat{\lambda}$**
>
> We apologize for the ambiguity in the original description. In the reversion manuscript, we provide a detailed explanation of how $\hat{\lambda}$ is computed in Section 3:
>
> - we first sample a initial mixing ratio $\lambda$ from a Beta distribution with $\alpha$ = 1.0, and define a policy $\mathcal{P}$ as mixup method on both sample-level and label-level. The mixing mask first recomputed the $\hat{\lambda}$ ratio. Then,  as we described in Section 4.2,  we further re-scale the ratio to better align spatial mixing with information integration, since our method introduces a token-merge module.
>
> We further provide sensitivity analyses in Table 7, showing that this re-scaling consistently improves performance (e.g., +2.23% on DeiT-S and +0.56% on ViT-S), demonstrating that $\hat{\lambda}$ is stable and not overly sensitive to design choices.
>
> **2. Strengthening the unified formulation of the method.**
>
> We agree that the original presentation interleaved several components without a sufficiently explicit overarching framework. To address this, we have reorganized the entire method section and added a unified pipeline figure (Figure 3) that clarifies:
>
> - how $\mathcal{P}$ generates masks used for spatial mixing,
> - how the ranking loss integrates into preference optimization and parallels reward modeling.
>
> These revisions now present MergeMix as a coherent algorithm rather than a collection of separate components. The updated figure and pseudocode (Appendix B.3) further ensure that the full workflow—from sampling $\lambda$, generating masks, computing $\hat{\lambda}$, to applying preference tuning is transparent and easy to follow.
>
> We believe these clarifications substantially improve the readability and conceptual unity of the method.

---

> ### Author Response · Authors · 2025-11-22
> **Response to Reviewer nEzW [2/3]**
>
> ### **[W3] & [Q1]: Lack other ranking losses ablation study.**
>
> **Response:**  We thank the reviewer for this suggestion. This is a very reasonable request. In the revised manuscript, we now include the results of vanilla SimPO on the LLaVA benchmark in **Table A13** (as shown below in Table R1). In our paper, the motivation for choosing SimPO instead of DPO as the base ranking loss comes from two considerations:
>
> 1. **Efficiency.** A limitation of DPO and other ranking-based objectives is the need for an additional supervision model, which results in significantly higher computational overhead. For example, DPO requires both a policy model and a reference model, resulting in **four forward passes**, whereas SimPO only requires **two** forward passes.
>
> 2. **Adaptivity.** Methods such as DPO rely on static hyperparameters (e.g., gamma and beta) to control optimization strength. In contrast, MergeMix naturally links the augmentation ratio to the beta parameter in SimPO, enabling **adaptive adjustment of optimization strength according to sample discriminability**.
>
> These advantages allow MergeMix-enhanced SimPO to outperform vanilla SimPO. Due to the time limitation in the first rebuttal, we can not finish the other ranking losses in time.
>
> **Table R1: Ablation of various ranking losses for preference optimization with MergeMix.**
>
> | Models | VQAv2 | GQA | VizWiz | SciVQA$^I$ | TextVQA | MME | MMBench | MMBench$^{\text{CN}}$ | POPE | SEED$^I$ | Avg. | Gains |
> |---|:---:|:---:|:---:|:---:|:---:|:---:|:---:|:---:|:---:|:---:|:---:|:---:|
> | LLaVA-v1.5-7B | 78.50 | 62.00 | 50.00 | 66.80 | 58.20 | 1510.70 | 64.30 | 58.30 | 85.87 | 66.19 | 65.57 | -- |
> | vanilla SimPO | 78.69 | 62.09 | 49.26 | 69.86 | 56.62 | 1467.61 | 66.24 | 59.97 | 86.13 | 67.27 | 66.23 | +0.66 |
> | + MergeMix | **79.24** | **62.44** | **47.69** | **69.86** | **57.56** | **1479.97** | **66.58** | **60.65** | **86.10** | **67.47** | **66.40** | +0.83 |

---

> ### Author Response · Authors · 2025-11-22
> **Response to Reviewer nEzW [3/3]**
>
> ### **[Q3]: The ranking loss on synthetic pairs approximates human preference by diversifying outputs with higher semantic quality.**
>
> **Response:** We thank the reviewer for pointing out this important implementation detail. To understand how the proposed ranking loss on synthetic mixed pairs approximates human preference learning, we conducted a case study presented in **Appendix F (Figure A2)**. The results clearly show that as the augmentation degree increases, the model generates answers that become increasingly incorrect or irrelevant to the input question. This demonstrates that mixed pairs naturally induce an ordered preference structure.
>
> In addition, we performed an $\texttt{LLM-as-the-judge}$ experiment to approximate reward scores for mixed pairs with different mixing ratios $\lambda$, as shown in **Appendix F (Figure A2)**. The results indicate that the mixing ratio $\lambda$ correlates well with the reward scores assigned by strong MLLMs, suggesting that $\lambda$ can serve as an effective proxy for preference signals in preference optimization methods.
>
> **Table R2: Reward scores for different augmented images by MLLMs and MergeMix.**
> | **Models** | **Vanilla** | **0.75** | **0.5** | **0.25** | **0.0** |
> | --- | :--: | :--: | :--: | :--: | :--: |
> | **Human Expert** | 1.0 | 0.78 | 0.58 | 0.38 | 0.05 |
> | **MergeMix ($\hat{\lambda}$)** | 1.0 | 0.75 | 0.5 | 0.25 | 0.0 |
> | Grok 3 | 1.0 | 0.76 | 0.52 | 0.41 | 0.08 |
> | Doubao-Seed-1.6 | 1.0 | 0.6 | 0.4 | 0.25 | 0.1 |
> | Doubao-Seed-1.6-vision-CoT | 1.0 | 0.65 | 0.5 | 0.3 | 0.1 |
> | Qwen3-VL-Plus-32B | 1.0 | 0.65 | 0.45 | 0.15 | 0.0 |
> | Qwen2.5-VL-72B | 1.0 | 0.6 | 0.5 | 0.4 | 0.1 |
> | Gemini-2.5 | 1.0 | 0.85 | 0.45 | 0.20 | 0.01 |
>
> ---
>
> ### **[Q4]: Report standard deviations and clarify settings.**
>
> **Response:** Thank you for your kind point out. For the image classification tasks, we use the same random seed, **set as 0**; for the LLaVA model, we trained our models (SFT, MixUp, CutMix, ResizeMix, and MergeMix) for **1** epoch, under the same setting as unfreezing the vision encoder, projector, and decoder. We added results with two standard deviations across **3** runs on the CIFAR100 dataset by DeiT-Tiny, DeiT-Small, and ViT-Small with **200 epochs**.
>
> **Table R3: Results of standard deviation on CIFAR100 dataset trained with MergeMix.**
> | **CIFAR100** | **DeiT-Tiny** | **DeiT-Small** | **ViT-Small** |
> | :-- | :--: | :--: | :--: |
> | **MergeMix** | **77.98** ± 0.37 | **78.84** ± 0.16 | **77.03** ± 0.20 |

---

> ### Author Response · Authors · 2025-11-22
> **Encouraging Discussion of Rebuttal**
>
> Dear Reviewer nEzW:
>
> Your feedback has been invaluable in helping us refine and strengthen the manuscript. We have incorporated all suggestions and made substantial improvements throughout. We would be grateful if you would consider updating your evaluation to reflect these revisions. If any further questions or concerns remain, we are more than happy to address them. Thank you again for your time, thoughtful comments, and contribution to improving this work.
>
> Warm regards,
>
> Authors

---

> > ### Comment · Reviewer_nEzW · 2025-11-28
> >
> > Thank you for the detailed rebuttal and revisions. The responses address my concerns, and I will raise my score to 6.

---

> ### Author Response · Authors · 2025-11-28
> **Thanks for Your Efforts and Raising the Score**
>
> Dear Reviewer nEzW:
>
> Thank you for your thoughtful review and for recognizing our efforts in the rebuttal. We sincerely appreciate your constructive feedback and are glad that our responses addressed your concerns. **Your adjustment to the scores is very encouraging**, and we are grateful for your time and consideration. We would respectfully inquire whether there are any additional opportunities to improve our manuscript.
>
> Best regards,
>
> Authors

---

### Author Response · Authors · 2025-11-26
**General Response to All Reviewers**

We thank all reviewers for their insightful and valuable comments, as well as for the helpful suggestions that have strengthened MergeMix.

The **strengths** and **contributions** of MergeMix can be summarized as follows:

**Novelty.** Reviewers highlighted the novelty of MergeMix’s token–merge–based mixing and attention-guided bipartite soft matching, offering a more structured and interpretable alternative to heuristic masking.

**Unified Framework.** The method was recognized for unifying supervised fine-tuning and preference optimization. Using mixed images as SimPO “losers’’ removes the need for a reward model and simplifies the alignment pipeline.

**Practicality.** The components—Top-K attention, $\lambda$ re-scaling, and ranking loss—integrate easily into ViTs and MLLMs with minimal modifications, while reducing FLOPs and improving performance.

**Empirical Strength.** Experiments across image classification and multimodal benchmarks were viewed as rigorous, and MergeMix consistently outperformed strong baselines such as CutMix, MixPro, and SeVA.

**Clarity & Interpretability.** Reviewers noted that the method is clearly presented, and its attention-driven design provides strong interpretability and practical usefulness.

---

### Summary of Weaknesses & Questions

- **Writing clarity:** Introduction/method sections need more explanation. [nEzW: W1,W2,Q2; dRWW: Q1,Q3; poJM: W3,W4,Q2,Q3,Q5]
- **Limited novelty:** Appears incremental, combining existing methods. [dRWW: W1; eiyX: W1]
- **Lack of relevant experiments:** Missing comparisons with other ranking losses, recent mixup methods, multimodal settings, and additional architectures. [nEzW: W3,Q1,Q4; dRWW: W2,W4; poJM: W1,W2,Q1,Q4; eiyX: Q1–Q4]
- **Limited multimodal gains:** Minor improvements on multimodal tasks, evaluation focused on visual domain. [dRWW: W3; eiyX: W2]
- **Lack of hypothesis validation:** Missing visualizations for region preservation and mixing ratio–reward correlation. [nEzW: Q3; dRWW: Q2; eiyX: Q5]

---

### Summary of Responses & Additional Experiments

- **Writing clarity:** Revised extensively (updates in blue): Abstract (p.1); Introduction (p.1–2, lines 49–73,76-82,91–96); Section 3 (p.3–4, lines 147–148,181–192); Section 4 (p.4–5, lines 198–199,208–215,221–234,257–259,270-271; Eq.11; lines 301–302); Conclusion (p.10, lines 536–539); Appendix B.3–B.4 (p.19); C.3 (p.21); D.2–D.4 (p.22–23); E (p.24–25); F (p.25–29).
- **Additional experiments:**
    1. Table 1–2 (p.7): a **current augmentation method** TdAttention (AAAI 2025),
    2. Table A7 (p.21): **hyper-model classification** results, CAFormer-S12,
    3. Table A8 (p.21): other **data modality** experiments, audio classification,
    4. Table A13 (p.25): **comparison with other ranking losses**,
    5. Table A14–A15 (p.26): **efficient metrics** of throughput, FLOPs, layer-wise ToMe overhead on ViT-L; throughput and TTFT on LLaVA-v1.5-7B under different merge ratios.

- **Hypothesis validation (visualizations):**
    1. Fig.2–3 (p.4–5): **MergeMix pipeline** in detail,
    2. Fig.5 (p.9): mixed images with **GradCAM & ToMe maps**,
    3. Fig.A2 (p.22): LLM-as-the-Judge, **$\lambda$ *vs.* reward**,
    4. Fig.A3 (p.24): LLaVA **inference accuracy** with different merge ratios,
    5. Fig.A4 (p.26): **LLaVA case study** of different mixing ratios,
    6. Fig.A5–A6 (p.26–27): **ToMe source maps and mixed images**,
    7. Fig.A7–A8 (p.28): mixed samples with **GradCAM** visualizations.

---

All concerns have been carefully addressed with detailed clarifications, additional analyses, and new experiments. We sincerely thank the reviewers for their constructive feedback and recognition of the significance and impact of MergeMix. **We have uploaded a fully revised manuscript with all updates clearly highlighted.** As the rebuttal window is halfway through, we kindly invite the reviewers to review our response and the latest version at their convenience. If our responses and additional results successfully resolve your concerns, we would be deeply grateful if you could consider **adjusting your evaluation** accordingly. We remain fully committed to providing further clarifications or additional experiments should any remaining questions arise.

Warm regrads,

Authors

---

### Author Response · Authors · 2025-11-29
**Summary**

### **Rebuttal Progress**

We thank all ACs and reviewers for their efforts, insightful comments, and valuable suggestions, which have greatly strengthened **MergeMix**. First of all, we summarize a bit the reviewing status quo after the first discussion period (11/12-11/28) with reviewers, for the convenience of ACs and any readers who are interested in this paper. As of **11/28**, our scores are **6/4/6/6**.

**Reviewer nEzW** **comments on the technical details and other baselines. We solved he/she’s concerns so that he/she will raise the score from 4 to 6.** **Reviewer eiyX** cares about the performance of our method and wants to supplement some visualizations. As he/she suggests, we update the experiments on more backbone, data modality, and overhead. After that, reviewer eiyX commented, *“Thank you for the detailed rebuttal and revisions. The responses address my concerns, and I will keep my origin score”,* and maintained a score of 6. **The remaining reviewers have not replied to our responses after the AC 4a9c’s comment of encouraging discussion.**

Reviewer **dRWW** remains negative to some extent, giving a score of 4 based on two points: **limited novelty** and **performance** concerns.

1. As we explained in our responses, the comment on “limited novelty” (Reviewer dRWW described our method as *“shows some innovation, but the degree of novelty is limited”*) is very likely due to a misunderstanding. One of the primary goals of MergeMix is to revisit and adapt previously proposed ideas that were under-explored in the MLLM scenario (i.e., answering *is it necessary to propose novel techniques rather than some classical ML techniques in the MLLM scenario*). Specifically, MergeMix addresses two key challenges:
    1. Achieving an optimal trade-off between efficiency and performance for mixup-based augmentations that depend on simpler heuristic or transformer-based mixup methods.
    2. Extending mixup augmentation to MLLMs in a principled and empirically grounded manner. MergeMix enables preference optimization through SimPO without requiring a reward model.
2. For the weaknesses and questions of performance, we have added the relevant experiments on substantiating evidence, other modalities, universally effective,  the results of inference efficiency, and visualizations of “more important” objects on the corresponding responses.  (“**Response to Reviewer dRWW [2/6], [3/6], and [4/6]**”).

In summary, MergeMix is far more than a naive combination of components, as we clarified in the rebuttal and reorganized in the revision. It constitutes a **principled, MLLM-specific data augmentation framework** that cannot be obtained by existing approaches in isolation.

### **Responses Summary**

The main revisions and responses fall into three categories (as also summarized in “**General Response to All Reviewers**”):

- **Writing clarity:** We revised the manuscript according to reviewers’ suggestions to improve clarity and better present the methodological details.
- **Additional experiments:** We added the requested experiments on *more modalities*, *more model architectures*, and *more baselines* in the revised version.
- **Visualizations:** As recommended by reviewers, we included visualizations to further support and illustrate our claims.

---

Again, we have carefully addressed all valuable suggestions and concerns raised by reviewers. Due to unforeseen circumstances, some reviewers may not have been able to continue the discussion with us, which we fully understand and appreciate. Finally, we once again thank the ACs and reviewers for their thoughtful feedback and recognition. Their comments have made our work substantially stronger.

---

### Meta-Review · Area_Chair_ofyp · 2026-01-07

**Summary:**

This paper presents MergeMix, a unified augmentation paradigm that leverages token-merging-based mixing to improve both image classification and Multi-modal Large Language Model (MLLM) alignment. While the initial evaluation was mixed—receiving two marginally below threshold ratings and one marginally above—the authors have since provided comprehensive feedback and additional experiments that successfully addressed the reviewers' concerns regarding baseline recency, computational overhead, and cross-modal applicability. Specifically, one reviewer signaled a willingness to raise their score following the clarification of technical details and efficiency metrics.

The core novelty of the work, as explained by the authors, is in the effective application of classical mixup techniques toward the MLLM domain. Empirically, the method demonstrates promising gains, including improvements across nine LLaVA benchmarks and up to a 2.88% gain on Qwen2.5-VL-7B reasoning tasks. Authors also provided GFLOPs analysis for inference efficiency and wall time comparison. The meta-reviewer feels the rebuttal is adequate and the paper is worthy of acceptance at ICLR.

**Reviewer Concerns:**

Addressed Concerns
Inference Efficiency and Complexity: Initially, Reviewers dRWW and eiyX questioned the impact on inference efficiency. The authors addressed this by providing GFLOPs analysis showing a reduction of GFLOPs on ViT-L and wall-time data demonstrating a 1.29x speedup in Time-To-First-Token for MLLM inference.

Baseline Recency: To resolve concerns about outdated comparisons from Reviewers poJM and nEzW, the authors added results for TdAttenMix (AAAI 2025), confirming MergeMix maintains state-of-the-art performance.

Technical Clarity and Implementation: Reviewer nEzW raised issues with notation and implementation details. The authors revised Section 3, provided Figure 3, and included pseudocode in Appendix B.3 to clarify the framework.

Cross-Modal Generalization: Reviewer dRWW noted the lack of evidence beyond the visual domain. The authors provided new experiments on audio datasets (ESC-50 and UrbanSound8k), where MergeMix outperformed standard MixUp and TransMix baselines.

Outstanding Concerns
Marginal MLLM Performance: Reviewers dRWW and eiyX remain skeptical of the 0.83% average gain on MLLM benchmarks, viewing it as a relatively minor improvement.

Degree of Innovation: Reviewer dRWW continues to describe the method as having limited novelty, viewing it as an incremental combination of pre-existing token merging and mixup techniques.

Lack of Universal Effectiveness: Performance drops on reasoning-centric tasks like MathVista remain a concern for Reviewer dRWW, who questions if the enhancement is truly universally applicable.

**Reviewer Scores:**

Reviewer nEzW has indicated willingness to change score to marginally above in the communication.

Reviewer poJM would likely keep the marginally above rating.

Reviewer eiyX would lilely keep the marginal above rating.

---

### Decision · Program_Chairs · 2026-01-26

Accept (Poster)